# Sub-nanometer depth resolution and single dopant visualization achieved by tilt-coupled multislice electron ptychography

Zehao Dong [1], Yang Zhang [1], Chun-Chien Chiu[2], Sicheng Lu[1], Jianbing Zhang [1], Yu-Chen Liu[2], Suya Liu[3], Jan-Chi Yang [2,4], Pu Yu [1], Yayu Wang [1,5,6] & Zhen Chen [7,8] ✉

Real-space, three-dimensional imaging of atomic structures in materials science is a critical yet challenging task. Although scanning transmission electron microscopy has achieved sub-angstrom lateral resolution through techniques like electron ptychography, depth resolution remains limited to only 2 to 3 nanometers using single-projection setups. Attaining better depth resolution often requires large sample tilt angles and numerous projections, as demonstrated in atomic electron tomography. Here, we introduce an extension of multislice electron ptychography, which couples only a few small-angle projections to improve depth resolution by more than threefold, reaching the sub-nanometer scale and potentially approaching the atomic level. This technique maintains high resolving power for both light and heavy atoms, significantly enhancing the detection of individual dopants. We experimentally demonstrate three-dimensional visualization of dilute praseodymium dopants in a brownmillerite oxide, $Ca_2Co_2O_5$, along with the accompanying lattice distortions. This approach can be implemented on widely available transmission electron microscopes equipped with hybrid pixel detectors, with data processing achievable using high-performance computing systems.

Visualizing the three-dimensional (3D) structure of materials, particularly dopants or vacancies within bulk samples, poses a significant challenge, yet is crucial for advancements in condensed matter physics[1], chemistry[2], and semiconductor industries[3,4]. Although aberration-corrected scanning transmission electron microscopy (STEM) has already achieved sub-angstrom lateral resolution[5,6], depth resolution along the beam direction remains significantly inferior with conventional optical depth-sectioning methods[7–9]. Electron tomography has attained atomic resolution in all three dimensions[10,11], but it requires numerous projections, large tilt angles, and offers a limited field of view[12]. Recent developments in multislice electron ptychography (MEP) have shown promising results, achieving depth resolution better than 3 nm in thick crystalline samples using only a single projection, while simultaneously resolving light elements such as O, B, and N[13–19].

Improving depth resolution in STEM can be accomplished by increasing the probe-forming semi-angle with advanced aberration correctors[20]. While this approach has yielded a depth resolution of 2.1 nm[21], it is restricted to thin samples due to limited depth of field and electron channeling effects[22]. By comparison, in diffractive imaging

---

[1]State Key Laboratory of Low Dimensional Quantum Physics, Department of Physics, Tsinghua University, Beijing, China. [2]Department of Physics, National Cheng Kung University, Tainan, Taiwan. [3]Shanghai Nanoport, ThermoFisher Scientific, Shanghai, China. [4]Center for Quantum Frontiers of Research & Technology (QFort), National Cheng Kung University, Tainan, Taiwan. [5]New Cornerstone Science Laboratory, Frontier Science Center for Quantum Information, Beijing, China. [6]Hefei National Laboratory, Hefei, China. [7]Beijing National Laboratory for Condensed Matter Physics, Institute of Physics, Chinese Academy of Sciences, Beijing, China. [8]School of Physical Sciences, University of Chinese Academy of Sciences, Beijing, China. ✉e-mail: zhen.chen@iphy.ac.cn

techniques like MEP, depth resolution is primarily governed by the maximum scattering angle captured by the detectors[23,24]. To address these limitations, we introduce tilt-coupled multislice electron ptychography (TCMEP), which employs a moderate probe-forming semi-angle combined with intentional off-axis sample tilting to capture higher-angle scattering information. Simulations show that the most significant improvement occurs at small tilt angles, with atomic-scale depth resolution achievable at tilt angles of approximately 4°, corroborating previous proof-of-principle demonstrations using similar reconstruction schemes[25,26]. Our experiments with TCMEP achieve sub-nanometer depth resolution, and successfully transfer higher-frequency information along the depth dimension through the sample tilt series. This improvement significantly enhances the identification of individual dopants and associated lattice distortions in three dimensions, thereby providing valuable insights into the physical properties of materials. TCMEP requires only small tilt angles (~4°) and a modest number of scans (fewer than 5 in this study), making it compatible with standard STEM instruments equipped with conventional double-tilt sample holders.

## Results

### Principle and reconstruction process

In TCMEP, we simultaneously reconstruct a shared object function using multiple four-dimensional STEM (4D-STEM) datasets. These datasets are acquired from the same region of the sample, with the specimen intentionally tilted by small angles—significantly smaller than 1 radian—away from the zone axis (Fig. 1a). A pre-reconstruction alignment procedure corrects relative shifts between datasets, followed by further refinement during the ptychographic reconstruction (details in Methods). A similar approach in X-ray ptychography has been shown to effectively improve depth resolution[27]. While beam tilt could serve as an alternative to specimen tilt, it introduces additional aberrations in the electron beam, complicating TCMEP reconstructions. In the following sections, we demonstrate how specimen tilts in electron ptychography unlock new imaging possibilities, especially by enhancing 3D resolution under relatively moderate illumination intensities, which have only been explored in proof-of-principle studies using simulated datasets[25,26].

To illustrate the advantages of TCMEP, we analyze the Fourier space representation of the information transferred in the reconstructed result (Fig. 1b). To make a direct comparison with conventional focal-series ADF imaging, the 3D information transfer via MEP from a single dataset is qualitatively modeled as a cone with an effective semi-angle, $\beta_{MEP}$. This angle generally depends on the maximum diffraction angle, but is also constrained by the probe's convergence angle under finite—and particularly low—dose conditions[16,19,28,29]. When the sample is tilted by an angle $\theta$ in real space, the corresponding information limit in Fourier space is similarly tilted. Coupling datasets acquired at tilt angles $\pm\theta$ fuses their information, resulting in an expanded cone of information transfer with an increased semi-angle, $\beta_{TCMEP}$. This process captures higher-spatial-frequency features along the $z$-axis, thereby improving depth resolution. Such improvement can also be interpreted through the concept of a tilted Ewald sphere as an alternative perspective[27].

The reconstruction process for TCMEP is illustrated in the flow-chart in Fig. 1c. Unlike conventional MEP, our algorithm operates in a parallel iterative manner, where $N$ (the number of datasets) 4D-STEM datasets collected at different tilt angles are coupled to reconstruct a unified multislice object. Given the relatively small tilt angles (≪1 rad), we approximate the tilted object model by introducing an interlayer shift. During each iteration, $N$ distinct probes (for each dataset) propagate through the $N$ tilted objects in parallel, and the exit waves are used to calculate the loss function, which guides updates for both the object and the probes. This iterative process continues until a convergent result is achieved. The phase of the final complex object

function represents the atomic potential in the sample, revealing the distribution of atomic defects[16,30], as detailed in our analyses below.

### Simulation on imaging a single dopant

We first evaluate the performance of TCMEP using simulations of a $SrTiO_3$ crystal containing artificially introduced interstitial and substitutional dopants. Figure 2a–c show reconstructed phase images at the same depth for three different maximum tilt angles—0°, 2°, and 4°—all under a constant total illumination dose of $2.5 \times 10^6 \ e/\text{Å}^2$. In each case, not only are all atomic columns resolved, but a faint feature corresponding to a single Sr dopant is visible as well (highlighted by circles). As the tilt angle increases, the visibility of the dopant improves due to reduced depth blurring, as demonstrated in the depth profiles in Fig. 2d–f. The phase-depth curves of the dopant (Fig. 2g) show a resolution-blurred Gaussian shape, which sharpens as the tilt angle increases. Similar trends are observed for lighter element dopants and vacancies, such as oxygen (Supplementary Fig. 1), highlighting the enhanced depth resolution achieved with TCMEP.

The Fourier transform of the reconstructed phase image reveals the boundary for 3D information transfer[19] (Fig. 2h, details in Supplementary Fig. 2), which agrees qualitatively with the schematic illustration in Fig. 1b, showing that the boundary expands along the depth dimension across all lateral spatial frequencies. The effective semi-angles $\beta_{MEP/TCMEP}$, defined by the slopes of the fitted quadratic curves near the origin, are measured at 207 mrad (11.8°), 352 mrad (20.1°), and 553 mrad (31.6°), respectively. Notably, $\beta_{MEP}$ in our simulation is approximately three times larger than the corresponding value from experimental results reported in ref. 19. This discrepancy primarily arises from the idealized simulation conditions, which do not account for experimental imperfections such as sample drift or partial spatial-temporal coherence. Moreover, the unavoidable roughness of the sample surface broadens the depth distribution in the Fourier spectra, leading to an underestimation of depth resolution in experimental results. Nevertheless, it is confirmed that the improvement in depth resolution is primarily driven by information gathered at higher angles.

To demonstrate TCMEP's dose efficiency, we performed simulations under varying illumination doses. Figure 2i presents depth sectioning images of the Sr dopant at total electron doses of $2.5 \times 10^4 \ e/\text{Å}^2$, $2.5 \times 10^6 \ e/\text{Å}^2$, and $2.5 \times 10^8 \ e/\text{Å}^2$. At the lowest dose ($2.5 \times 10^4 \ e/\text{Å}^2$), the dopant is barely distinguishable using conventional MEP. However, even a slight tilt of 2° renders it discernible, and the contrast further improves at a 4° tilt. Depth blurring is also significantly reduced at higher tilt angles. At higher doses ($2.5 \times 10^6 \ e/\text{Å}^2$ and $2.5 \times 10^8 \ e/\text{Å}^2$), nearly identical results are observed, both showing substantial improvements in depth resolution. Our simulations indicate that the typical experimental dose (~$10^6 \ e/\text{Å}^2$) is sufficient to achieve atomic-scale depth resolution (~4 Å) with TCMEP using only five tilts up to 4°. Notably, at doses around $10^3 \ e/\text{Å}^2$, dopant atoms become nearly indistinguishable from artifacts under current imaging conditions.

We fit all phase-depth curves with a Gaussian function $y = A \exp\left[-\frac{(x-\mu)^2}{2\sigma^2}\right] + B$, determining depth resolution by the full width at 80% of the maximum (FW80M, $d = 1.33\sigma$), following the method in prior work[16]. As shown in Fig. 2j, with a large illumination dose and a maximum tilt of 4°, the depth resolution improves by a factor of 2 to 3, achieving nearly atomic resolution (~0.45 nm). At lower doses, with comparable tilt angles, the improvement is even more pronounced, reaching around 0.62 nm—approximately 4 times better than MEP (2.33 nm). This improvement arises because electrons in the bright-field disk contribute significantly to the 4D-STEM dataset, while electrons in the dark-field regions remain below the Poisson noise level at low doses. TCMEP incorporates higher-angle information into the bright-field disk through specimen tilts, thereby significantly improving dose efficiency in depth sectioning.

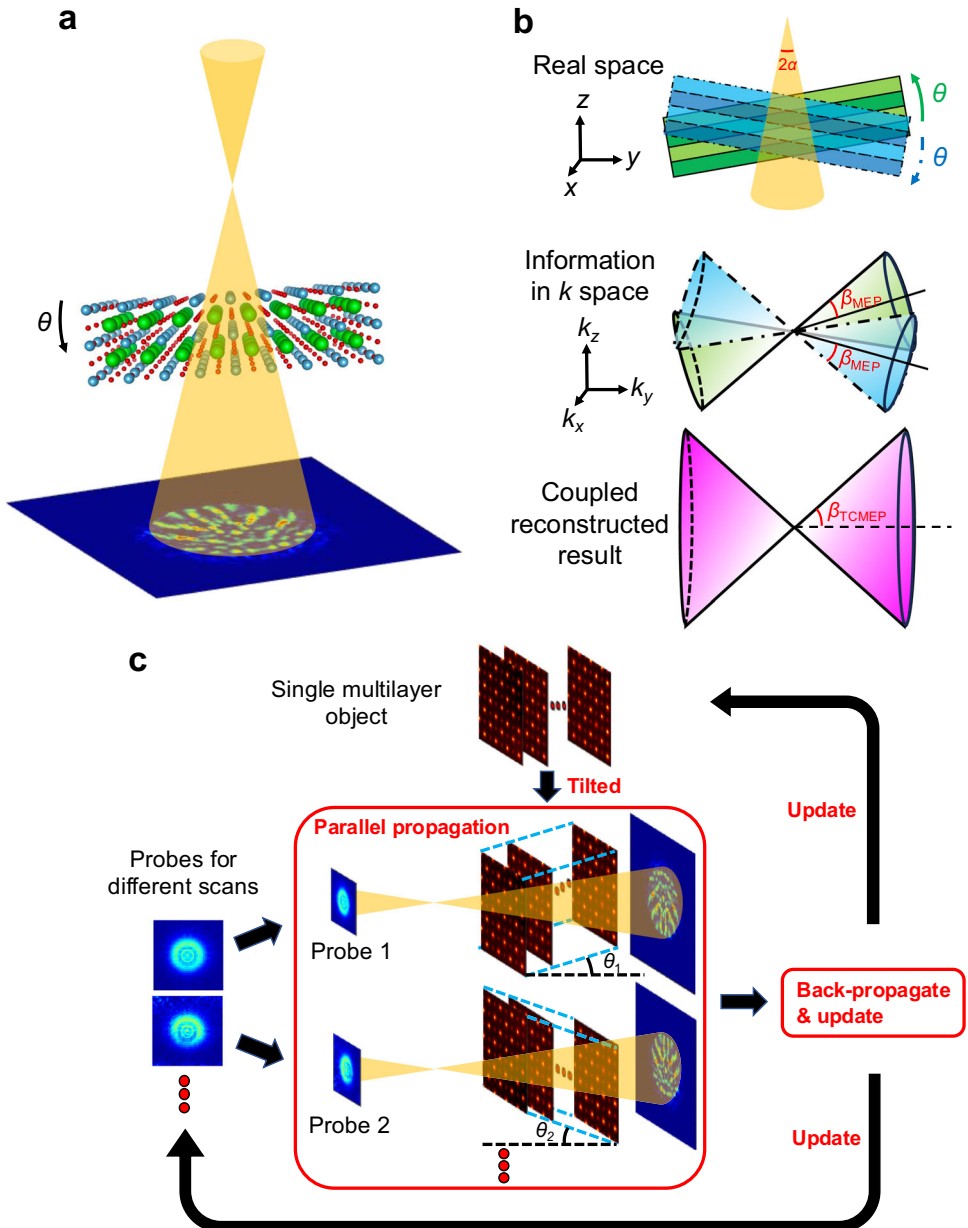

**Fig. 1 | Principle and process of tilt-coupled multislice electron ptychography (TCMEP). a** Schematic of the experimental setup for TCMEP, where the sample is tilted by a small angle $\theta$ relative to the zone axis. Multiple datasets are acquired under different tilt conditions during the experiment. **b** Illustration of how TCMEP enhances depth resolution. The upper panel shows the sample tilted by angle $\theta$ from the zone axis. The middle panel depicts the corresponding information transfer in Fourier space, represented as a cone with an effective semi-angle $\beta_{MEP}$ for 3D information transfer. The lower panel demonstrates how coupling datasets from different tilts broadens the information transfer limit, effectively increasing the semi-angle to $\beta_{TCMEP}$, thereby improving depth resolution. **c** Flowchart of the TCMEP reconstruction algorithm, involving parallel processing of all datasets to optimize a single multilayer object.

## Depth resolution to sub-nanometers

We next present experimental results on a twisted bilayer $SrTiO_3$ sample with a relative twist angle of approximately 9°. Twisted bilayer systems are known for their exotic quantum many-body phenomena and potential applications in twistronics[31–33]. Depth-resolving techniques in STEM are essential for probing the buried interfaces in such samples[18]. Our fabricated sample exhibits a clean, sharp interface, with each layer being a few nanometers thick. In the projection (Fig. 3c), a Moiré pattern emerges due to the intermixing of the top and bottom layers, which should be clearly separated by ideal depth sectioning. Therefore, this system serves as a useful benchmark for assessing depth resolution by tracking the residual intermixing near the interface.

Figure 3a, b present real-space images and their corresponding Fourier transforms (FFTs), showing five slices (each 4 Å thick) from two reconstructions using MEP and TCMEP (see Supplementary Figs. 3 and 4 for all slices). In the first slice, only the top $SrTiO_3$ layer is resolved in both reconstructions, as indicated by the dashed circles in the FFTs. A key distinction lies in the Moiré pattern's extent along the depth dimension. In the TCMEP results, the Moiré pattern appears and fades more rapidly along the z-axis compared to MEP, where it diminishes more gradually. In the final slice, the residual Moiré pattern remains in the MEP reconstruction, whereas TCMEP resolves only the bottom $SrTiO_3$ layer, demonstrating superior layer separation.

The difference in averaged phase-depth curves further confirms the enhanced depth resolution at higher tilt angles (Fig. 3d). By fitting

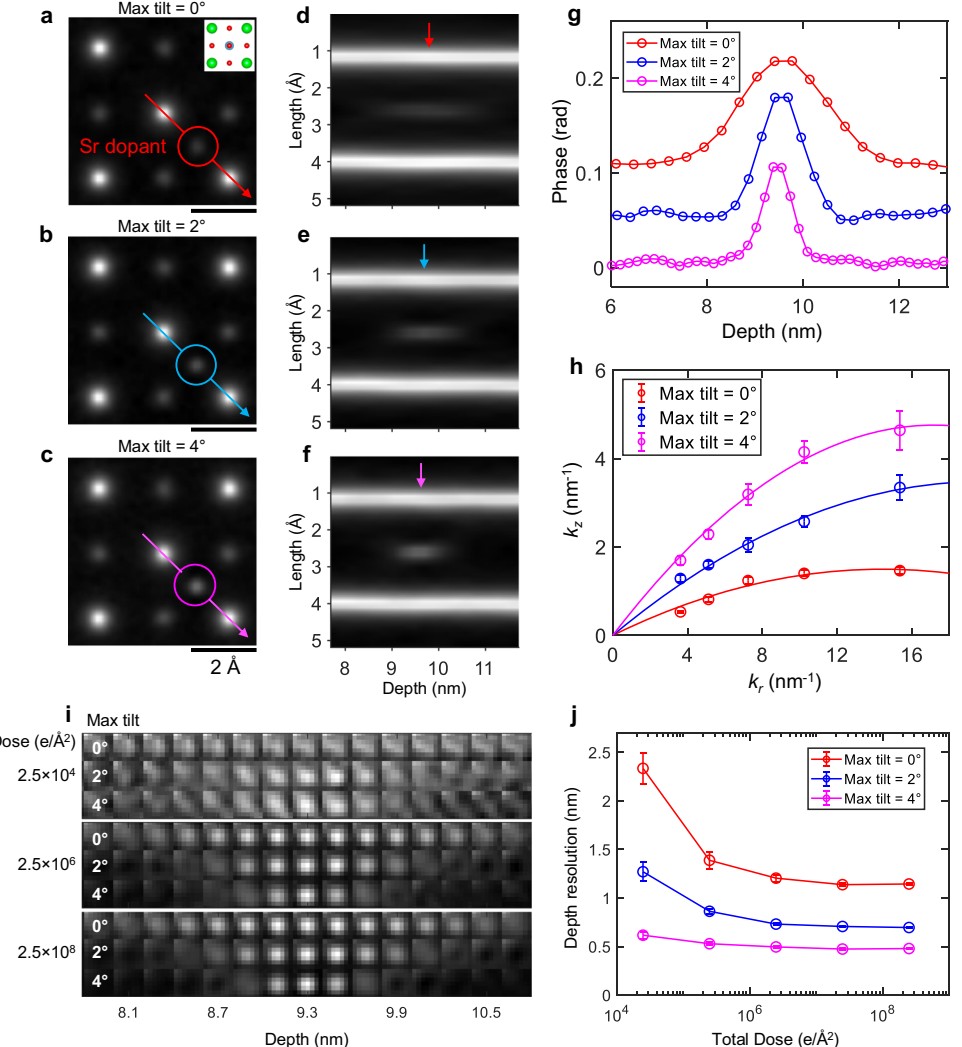

**Fig. 2 | Simulated imaging of a single Sr dopant using TCMEP. a–c** Reconstructed phase images of a slice containing a Sr dopant (marked by circles) under tilt conditions: 0° (**a**); 0° and ±2° (**b**); 0°, ±2°, and ±4° (**c**). The total illumination dose is $2.5 \times 10^6 \, e/\text{Å}^2$. The inset in (**a**) displays the structural model of the slice. Scale bars, 2 Å. **d–f** Depth profiles along the arrows in (**a–c**), illustrating the depth blurring of the Sr dopant. **g** Phase-depth curves for the Sr dopant under the three tilt conditions, with vertical offsets for clarity. **h** 3D information transfer boundary as a

function of lateral spatial frequency, fitted with a quadratic function passing through the origin. **i** Slice images of the Sr dopant at different depths (each slice 2.1 Å thick), arranged horizontally. Results for different tilt angles and illumination doses are stacked vertically. **j** Depth resolution obtained under different simulation conditions, plotted as a function of total illumination dose. Error bars are derived from residuals of curve fitting.

the curve for a 2° tilt angle, we achieve a depth resolution of approximately 0.9 nm at the interface, using a total dose of $8.4 \times 10^5 \, e/$ Å² (details in Supplementary Fig. 5). This resolution surpasses the chromatic aberration limit (-1.4 nm) and aligns well with our simulations under comparable conditions. Moreover, the corresponding Fourier analysis (Fig. 3e, details in Supplementary Fig. 6) reveals a universally improved depth resolution across all lateral spatial frequencies as the maximum tilt angle increases, consistent with the findings in Fig. 2h for the simulated datasets.

## Imaging dopant atoms and 3D lattice distortions

TCMEP is then applied to image dopant atoms, which play a critical role in modulating emergent phenomena within quantum materials[1]. Previous studies on cobalt oxides in the brownmillerite phase have demonstrated a tunable magnetic ground state upon doping[34]. Our experiments focus on a specific brownmillerite $(Pr_xCa_{1-x})_2Co_2O_5$ thin film (nominal $x$ is about 0.05) grown on a LaAlO₃ (001) substrate (Fig. 4a), composed of alternating stacks of $CoO_6$ octahedra and $CoO_4$ tetrahedra[35,36]. The unique advantage of this material lies in the

spontaneous breaking of lattice inversion symmetry after Pr doping, as evidenced by electron energy-loss spectroscopic (EELS) mapping in Fig. 4b, c. Pr substitutional atoms preferentially occupy the Ca2 sites (indicated by red arrows) over the Ca1 sites (blue arrows), despite the equivalence of these sites in the undoped parent phase. This selective substitution could be related to the spontaneous polar distortions observed in similar compounds[36]. Consequently, precise identification of Pr dopants using MEP or TCMEP requires distinguishing between these inequivalent Ca sites. This differentiation is critical to isolate true atomic substitutions from confounding factors such as intrinsic phase fluctuations, imaging artifacts, or beam-induced sample damage.

We use only three sample tilts at 0° and ±1° to reduce the experimental workload. Figure 4d and g show the projected phase images reconstructed with MEP and TCMEP, both achieving similar lateral resolution of around 0.4 Å. To identify Pr substitution in the Ca columns, depth sectioning was conducted along the Ca1 (blue arrow) and Ca2 (red arrow) rows (Fig. 4e and h, depth profiles for all Ca rows are provided in Supplementary Figs. 7 and 8). A clear distinction is observed between Ca1 and Ca2 in both reconstructions, with Ca1

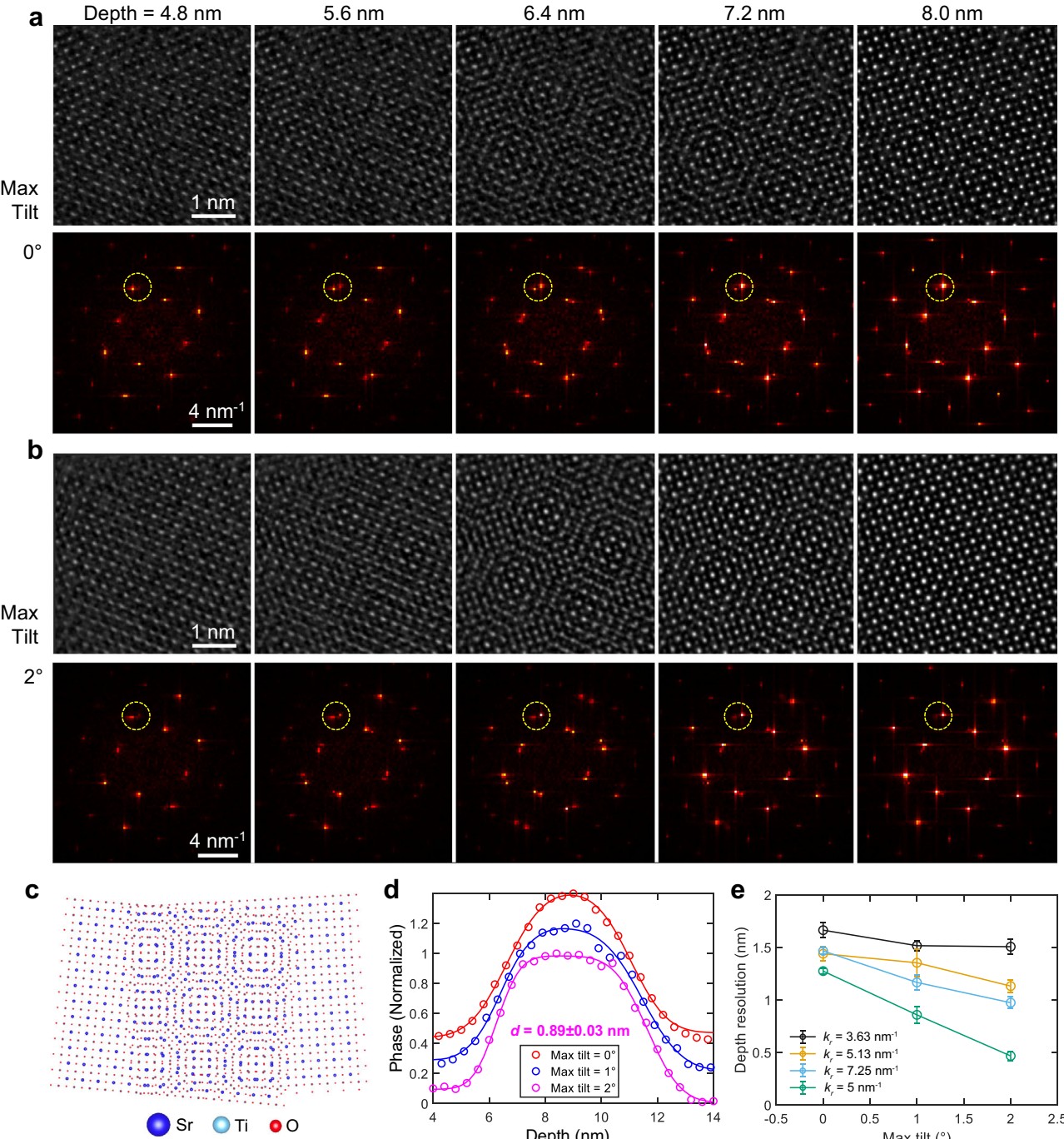

**Fig. 3 | Experimental validation of improved depth resolution using TCMEP.** **a**, **b** MEP (**a**) and TCMEP (**b**) reconstructed phase images from different depths (upper panels) with corresponding Fourier transforms (lower panels). Bragg peaks (yellow dashed circles) demonstrate the improvement in depth resolution. Scale bars, 1 nm (upper panels) and 4 nm⁻¹ (lower panels). **c** Structural model of the twisted bilayer SrTiO₃. **d** Phase profiles averaged from Sr and Ti columns in the bottom layer, fitted with Gaussian error functions (see Supplementary Fig. 5). TCMEP with a maximum tilt angle of 2° yields a depth resolution of approximately 0.9 nm at the interface. **e** Depth resolution as a function of maximum tilt angle for different lateral frequencies. Error bars are derived from residuals of curve fitting.

showing a uniformly distributed phase, while Ca2 exhibits randomly distributed additional phase peaks. These peaks result from the increased average atomic number ($Z$) when Pr atoms ($Z = 59$) replace Ca atoms ($Z = 20$). Since the phase value approximately scales with $Z^{0.67}$, the phase associated with Pr columns is expected to be roughly double that of Ca columns with the same atomic density. However, due to partial Pr substitution and depth-resolution-induced broadening, the observed phase enhancement from Pr is only approximately 20% in

our experimental results. Importantly, multiple dopant atoms can be detected within the same atomic column using TCMEP.

We then compare the depth profiles shown in Fig. 4e and h. Both reconstructions show similar phase distributions for the Ca2 rows, confirming that TCMEP does not introduce extrinsic artifacts or obscure intrinsic features. A key difference appears in the second Ca2 column (marked by black arrows), where TCMEP successfully resolves two adjacent peaks separated by approximately 4 nm in depth. This

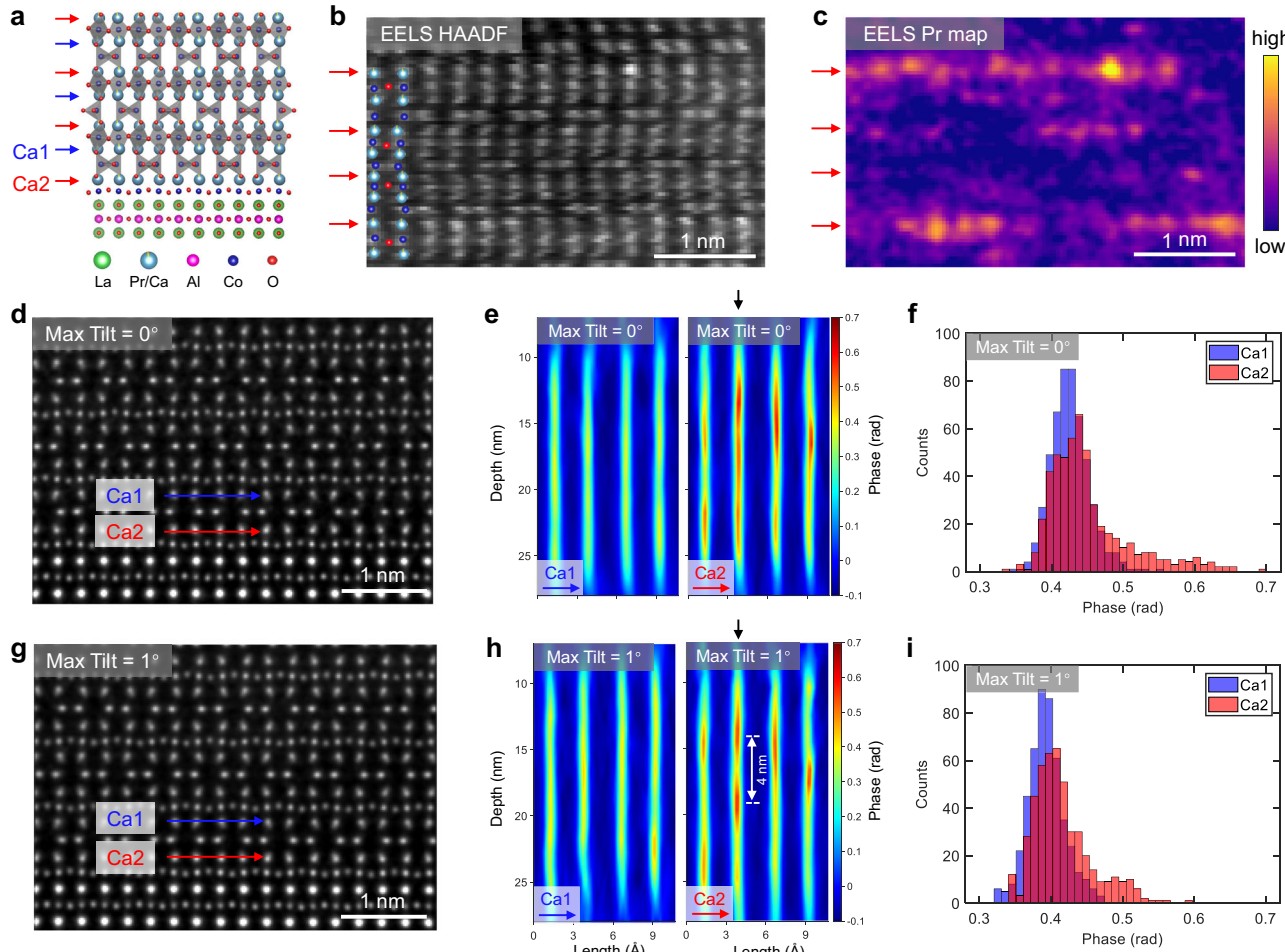

**Fig. 4 | Experimental imaging of Pr dopants in the $(Pr_{0.05}Ca_{0.95})_2Co_2O_5$ thin film. a** Structural model of the $(Pr_{0.05}Ca_{0.95})Co_2O_5$ film grown on a LaAlO₃ substrate. Two distinct Ca atom rows are marked by blue (Ca1) and red (Ca2) arrows, respectively. **b, c** Simultaneously acquired high-angle annular dark-field (STEM-HAADF) image (**b**) and Pr $M_5$-edge intensity map from STEM-EELS (**c**). The Ca2 rows are marked by red arrows, as identified in the HAADF image. **d** Projected phase image reconstructed using MEP without sample tilting. **e** Depth profiles

along the arrows in (**c**) for Ca1 (left) and Ca2 (right). **f** Statistical results of phase values for Ca1 and Ca2 sites from MEP results. **g** Projected phase image of the same region reconstructed using TCMEP with a maximum tilt of 1°. **h** Depth profiles corresponding to (**e**) for Ca1 (left) and Ca2 (right). A vertical black arrow highlights a 4 nm depth difference between two adjacent peaks in the Ca2 column, demonstrating the enhanced depth resolution achieved with TCMEP. **i** Statistical results of phase values for Ca1 and Ca2 sites from TCMEP results. Scale bars, 1 nm.

feature is obscured in the MEP reconstruction, due to its inferior depth resolution. Additionally, statistical analysis of phase values for the Ca2 sites reveals an extra shoulder in the TCMEP results (Fig. 4i), which broadens into an extended tail in the MEP reconstructions (Fig. 4f). These findings underscore TCMEP's superior depth resolution for identifying finer structures along the depth axis.

As a result, nearly all Pr dopants are clearly resolved in three dimensions using TCMEP, allowing for a detailed examination of the associated lattice distortions. Figure 5a displays the projected phase image reconstructed with TCMEP, using a customized colormap to emphasize phase variations at the Ca sites. We focus on the regions outlined by dashed rectangles, with corresponding depth profiles shown in Fig. 5b, c. In Fig. 5b, the Ca2 column reveals a Pr dopant centered at approximately 14 nm in depth, while the adjacent Ca1 site at the same depth is displaced away from the dopant. A similar pattern is observed in Fig. 5c, with a Pr dopant at around 18 nm in depth. Figure 5d, e provide in-plane views of the slice containing the Pr dopants, where phase maps for Ca columns are superimposed with atomic displacement maps relative to the projected phase image (atomic displacement maps from other slices exhibiting clear atomic structures are shown in Supplementary Movie 1). The Pr-doped Ca2 sites appear in yellow, confirming a higher Pr concentration in

these columns, consistent with the conclusions drawn from Fig. 4. Notably, the atomic displacements near the Pr dopants are larger, especially in the highlighted regions. On average, these displacements range from 5 to 10 pm and are generally directed away from the Pr dopants. Two mechanisms may be relevant for these distortions: (i) modification of Co-O bond lengths due to electron doping on Co ions[37], and (ii) the ionic radius mismatch between Pr³⁺ and Ca²⁺ ions. A thorough analysis of the correlation between dopant distribution and lattice distortions will be crucial for understanding the complex magnetic ground state of cobaltates under carrier doping.

## Discussion

We have demonstrated that TCMEP significantly enhances depth resolution and the visibility of single dopants, providing a powerful tool for visualizing the distribution of atomic defects. This advancement holds the potential to unlock insights into the physical properties of a wide range of materials, from semiconductor devices[3] to high-temperature superconductors[38]. The improved depth resolution to sub-nanometer scales also enables new capabilities for resolving complex 3D structures, such as nitrogen-vacancy centers[39], topological polar textures[40], and nanoscale phase separations[41].

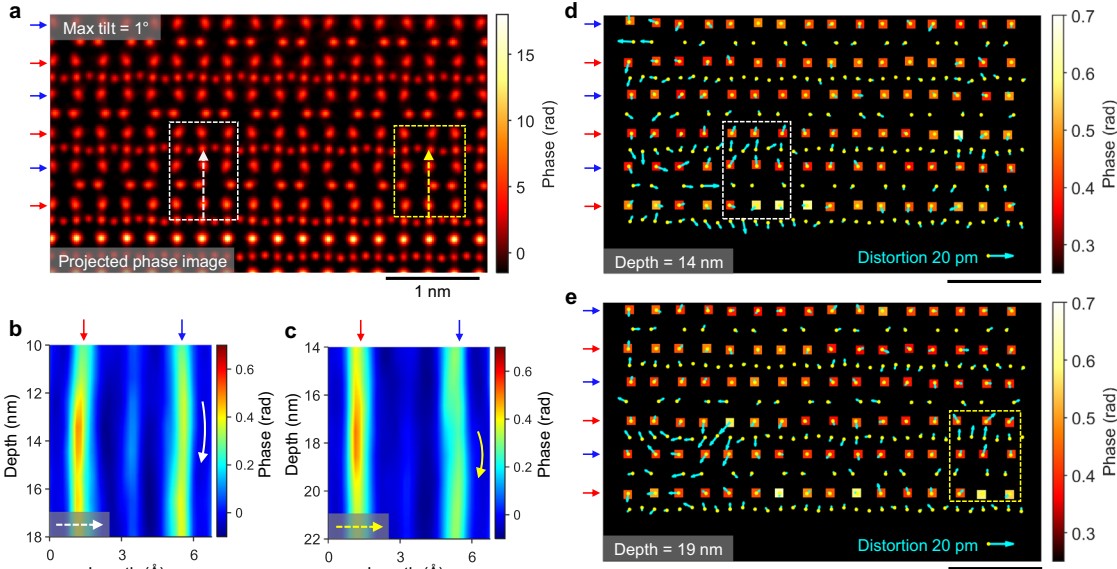

**Fig. 5 | Pr dopants and associated lattice distortions. a** Projected phase image reconstructed using TCMEP with a maximum tilt of 1°. Scale bar, 1 nm. **b, c** Depth profiles along the white (**b**) and yellow (**c**) dashed arrows in (**a**), showing Pr dopants located at depths of 14 nm (**b**) and 18 nm (**c**) within the Ca2 columns. Curved arrows indicate the associated distortions in the Ca1 columns at similar depths. **d, e** Phase maps of all Ca atoms for object slices at 14 nm (**d**) and 19 nm (**e**). Cyan arrows indicate atomic displacements relative to the projected phase image in (**a**). The white and yellow dashed rectangles mark regions corresponding to (**a**). A reference distortion magnitude of 20 pm is provided for comparison.

Depth resolution in MEP-based techniques is also influenced by the efficiency in retrieving phase information from convergent-beam diffraction patterns. For example, MEP reconstruction on simulated datasets for a weakly scattering $SrTiO_3$ nanoparticle (Supplementary Fig. 9) demonstrates a superior depth resolution (0.73 nm) compared to that for a strongly scattering $SrTiO_3$ crystal (Fig. 2) under identical parameters (1.20 nm). This improvement can be attributed primarily to the more interpretable diffraction patterns produced by nanoparticles, as the multiple scattering effects are significantly reduced (Supplementary Fig. 9b, c), thereby facilitating three-dimensional phase retrieval. Notably, even in this weak-scattering context, TCMEP enhances depth resolution from 0.73 nm (maximum tilt 0°) to 0.41 nm (maximum tilt 4°), indicating its broad applicability across different scattering regimes.

Although TCMEP's reliance on small tilt angles for the interlayer shift approximation may seem restrictive at first glance, we demonstrate that reconstructions remain both feasible and reliable even at a maximum tilt angle of 10° (~0.17 rad), well within the small-angle regime[42]. This approach achieves a depth resolution surpassing 3 Å (Supplementary Fig. 10), consistent with previous simulation results using analogous methodologies[25,26]. Notably, TCMEP can be extended to larger tilt angles with the implementation of projection algorithms used in tomography, as demonstrated by previous simulation studies[43–46]. Several experimental reports using sequential (rather than joint) reconstructions that combine MEP and tomography have achieved a 3D resolution better than 2.0 Å and a precision of 17 pm, utilizing 36 projections with a maximum tilt angle of 63°[47,48]. However, light atoms such as oxygen were unresolved in these studies due to uncertainties in scan positions and sample tilt registrations. In contrast, TCMEP significantly simplifies data acquisition and processing, reducing the overall experimental complexity and workload. As shown above, we used only 4 projections with a maximum tilt angle of 2° to achieve a depth resolution of approximately 9 Å and a lateral resolution better than 0.4 Å. Despite a slight trade-off in depth resolution (compared to ref. 48), our results demonstrate deep sub-angstrom lateral resolution, picometer-scale precision, and marked sensitivity to light atoms.

In summary, by introducing sample tilt-series into multislice electron ptychography, we capture information from higher angles and improve the experimental depth resolution to just a few angstroms. This method excels in imaging single dopants and atomic displacements in all three dimensions while remaining dose-efficient and requiring only a few tilts, making it compatible with the conventional aberration-corrected STEM instruments and double-tilt sample holders readily available in most laboratories. With further development, TCMEP could enable three-dimensional atomic resolution in the future.

Note added in proof: During the review process of this manuscript, we noticed another work by Schloz et al. [49], which proposed a defocus-series strategy to improve the three-dimensional reconstruction of multislice electron ptychography.

## Methods

### Sample growth and preparation for TEM measurements

Brownmillerite $(Pr_{0.05}Ca_{0.95})_2Co_2O_5$ thin films were grown by a customized pulsed laser deposition (PLD) system, at 640 °C with an oxygen pressure of 0.06 mbar. The laser (KrF, $\lambda = 248$ nm) energy density was set at 1.1 J/cm² with the repetition rate of 5 Hz. After the growth, the samples were cooled down to room temperature at a cooling rate of 10 °C/min at 0.06 mbar oxygen pressure. The crystalline structures of thin films were characterized by a high-resolution four-circle X-ray diffractometer (Smartlab, Rigaku) using monochromatic Cu $K_{\alpha1}$ radiation ($\lambda = 1.5406$ Å). TEM samples were prepared using a focused ion beam (FIB) instrument (Zeiss Auriga). The samples were thinned down to 100 nm using an accelerating voltage of 30 kV with a decreasing current from 240 pA to 50 pA, followed by a fine polish with an accelerating voltage of 5 kV with a current of 20 pA.

The freestanding $SrTiO_3$ (STO) films were grown by PLD, using KrF (248 nm) excimer laser. The STO thin films were deposited on (La,Sr)MnO₃ (LSMO) buffered (001)-oriented STO substrates, carried out under an oxygen pressure of 100 mTorr at a temperature of 700 °C, utilizing a laser power of 250 mJ and a laser repetition rate of 10 Hz. The heterostructure was then immersed in hydrochloric acid to dissolve the LSMO sacrificial layer and to separate STO film from the

single crystal substrate. The freestanding STO was then transferred onto another single-crystalline STO heterostructure grown on LSMO buffered (001)-oriented STO substrate, having a twist angle with respect to the STO single-crystal substrate. Thereafter, the sample was immersed in hydrochloric acid once again to obtain the freestanding twisted bilayer STO membrane. The STO membrane was then transferred onto a copper TEM grid. Sufficient rinsing was performed to make the interface of the two layers clean without any residue, as illustrated in the ptychographic reconstruction in Fig. 3 and Supplementary Figs. 3 and 4.

## STEM-EELS experiment

STEM-EELS experiments were performed on an aberration-corrected FEI-Titan Cubed Themis 60–300 TEM operating at 300 keV. EELS data and HAADF signal were simultaneously acquired with a beam current of 50 pA, a convergence semi-angle of 25 mrad, a GIF collection semi-angle of 56.5 mrad, a scanning step size of 0.35 Å, and a dwell time of 40 ms. The energy resolution was 0.8 eV. The energy dispersion was 0.05 eV/channel, under dual EELS mode. The pre-edge background was subtracted and the plural scattering was removed using a Fourier-ratio method. Sample drift was corrected using the reference structure from HAADF image by a custom script. Pr elemental map was obtained using the integrated intensity of Pr-$M_{4,5}$ edges around 931 eV and 951 eV.

## 4D-STEM experiments

Experiments on twisted SrTiO$_3$ bilayers were performed using an aberration-corrected Spectra 300 (Thermo Fisher Scientific) electron microscope, operating at 300 keV and equipped with an ultrahigh brightness Cold Field Emission Gun (X-CFEG). For the $(Pr_{0.05}Ca_{0.95})_2Co_2O_5$ thin film, a Titan Cubed Themis 60–300 (Thermo Fisher Scientific) electron microscope, also operating at 300 keV, was employed, featuring a high-brightness Schottky Field Emission Gun. A probe-forming semi-angle of 25 mrad was used throughout the experiments, and four-dimensional STEM (4D-STEM) datasets were acquired using an Electron Microscope Pixel Array Detector (EMPAD) with $124 \times 124$ pixels. The focal point of the probe was positioned approximately 20 nm above the sample surface. Each 4D-STEM dataset covered a scanning area of $9.3 \times 9.3$ nm$^2$, with $200 \times 200$ uniformly distributed scanning points, and an exposure time of 1 ms per diffraction pattern. The beam current was set to 30 pA for the $(Pr_{0.05}Ca_{0.95})_2Co_2O_5$ film (corresponding to a dose of $9.0 \times 10^5$ $e$/Å$^2$), but reduced to 7 pA for the SrTiO$_3$ sample to minimize beam damage (corresponding to a dose of $2.1 \times 10^5$ $e$/Å$^2$).

For the twisted bilayer SrTiO$_3$ experiments, tilt angles of ±2° and ±1° were acquired, and we respectively utilized 1 scan (−1°, max tilt 0°), 2 scans (±1°, max tilt 1°), and 4 scans (±1° and ±2°, max tilt 2°) for MEP or TCMEP reconstructions. In the $(Pr_{0.05}Ca_{0.95})_2Co_2O_5$ experiments, tilt angles of ±1° and 0° (from the [100] zone axis) were acquired, and we respectively utilized 1 scan (0°) and 3 scans (±1°, 0°) for MEP or TCMEP reconstructions. The depth resolution was around 2.1 nm for MEP and 1.5 nm for TCMEP (Supplementary Fig. 11). This enhanced depth resolution surpasses the improvements achievable through a simple threefold increase in illumination dose (from $9.0 \times 10^5$ $e$/Å$^2$ to $2.7 \times 10^6$ $e$/Å$^2$), as demonstrated by the simulations in Fig. 2j. In our experiments, exact tilt angles were not required, as further refinements were performed during the reconstruction process.

## Alignment of datasets and reconstruction process

Prior to 4D-STEM data acquisition, the probe was focused on a fixed spot near the region of interest (ROI) for about 15 minutes. As illustrated in Supplementary Fig. 12a, this resulted in the formation of a distinct structural defect, which was later used to relocate the ROI after tilting the sample. Alignment among those 4D-STEM datasets was performed before the TCMEP reconstruction. Initially, a conventional MEP reconstruction was performed on each dataset to determine the reference defect's exact positions (Supplementary Fig. 12a–c). All datasets were then aligned accordingly and cropped into a smaller overlapped region of $100 \times 100$ scanning points (individual MEP reconstructions shown in Supplementary Fig. 12d–f), on which the TCMEP reconstruction was performed. The slight residual misalignments will be further refined during the TCMEP iterations.

In the TCMEP reconstruction process, sample tilts were modeled by shifting the object functions, although a tilted Fresnel propagation function could serve as an alternative approach[50]. A Gaussian kernel with width $\sigma_{PSF}$ was applied to the calculated diffraction patterns to model the point spread function of detectors. A drift-correction algorithm was employed to refine the precise scanning positions from each dataset during TCMEP reconstruction[51]. Partial spatial coherence was accounted for using the mixed-state algorithm[52]. The update direction and step size ($\beta_{LSQ}$) for each iteration were determined through the least-squares maximum likelihood (LSQML) method[53–55]. Results of the mixed-state probes, probe position refinements, and tilt angle corrections are presented in Supplementary Fig. 13. Bayesian optimization for hyperparameter refinement[56] was not applied in this study, as conventional TCMEP reconstruction parameters, listed in Supplementary Table 1, led to sufficiently high convergence, consistent with previous report for MEP[16]. Smaller regularization factors[57] were used exclusively for TCMEP, as MEP does not converge well with equivalent factors due to its inferior depth resolution.

TCMEP reconstruction is computationally intensive and requires large GPU memory. All numerical processing in this study was performed using an Nvidia A100 GPU with 80 GB of memory. The parameters in Supplementary Table 1 were carefully selected to balance memory requirements with total reconstruction time based on our current implementation. For example, TCMEP reconstruction for twisted SrTiO$_3$ with a maximum tilt of 2° required approximately 10 GB of GPU memory and 30 h of computation time. Therefore, advancements in both hardware capabilities and algorithmic efficiency are crucial to accelerate the computational process for larger datasets, especially when striving for the ultimate atomic-scale depth resolution.

## 4D-STEM simulation

The simulated 4D-STEM datasets were all generated at 300 keV beam energy using the µSTEM software[58]. The probe's convergent semi-angle was 25 mrad, and was overfocused by 20 nm above the sample surface. $26 \times 26$ diffraction patterns with a 0.60 Å step were generated at different sample tilt angles. We used a 12.5 nm-thick SrTiO$_3$ structural model along the [001] zone axis with different artificially introduced dopants. The structural model contained $4 \times 4$ unit cells in real space ($15.6 \times 15.6$ Å$^2$ in area), and was sampled with $128 \times 128$ pixels. The corresponding reciprocal space sampling was 0.0641 Å$^{-1}$ per pixel, with $128 \times 128$ pixels and a maximum scattering angle of around 80 mrad. Lattice vibrations were simulated by a frozen-phonon model with 40 configurations. Poisson noise was incorporated to account for the finite illumination dose. Spatial and temporal incoherence was not considered since the simulations were only used to demonstrate qualitative tendencies. For each TCMEP simulation with multiple datasets, the interval of tilt angle was taken as 2°. The slice thickness for each reconstruction was chosen to be less than one-third of the depth resolution. This ensured that the slice thickness remained below the Nyquist sampling rate, preventing any alteration of the reconstructed results. Reconstructions of simulated datasets used a regularization factor of 0.1 in the depth dimension, with other parameters specified in Supplementary Table 1.

## Precision in measuring 3D atomic positions

The precision in measuring 3D position of atoms is tolerable with TCMEP, with an accuracy of 1.8 pm in plane and 0.3 nm in depth, estimated from the peak positions of Sr/Ti atoms (Supplementary Fig. 14). These values are both comparable to MEP results. Notably, the

in-plane precision of TCMEP (1.8 pm) is slightly worse than that of MEP (1.3 pm), partly due to slight misalignments among datasets and the additional position correction process. It should also be noted that the estimation for precision in the depth dimension is only an upper limit, due to the inevitable surface roughness and local curvature of the interface beyond the measurement uncertainty. Moreover, the statistical variation of 0.3 nm in depth indicates that the surface roughness of the fabricated $SrTiO_3$ sample is smaller than the size of a single unit cell (0.4 nm).

## Data availability
The 4D-STEM data presented in this study are available in Zenodo[59].

## Code availability
The code for TCMEP is available in Zenodo[59].

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

## Acknowledgements

This work was supported by the National Key Research and Development Program of China (MOST) (Grant No. 2023YFA1406400 and No. 2022YFA1405100), the National Natural Science Foundation of China (Grant No. U22A6005, No. 52273227, and No. 52025024), the Basic Science Center Project of NSFC (No. 52388201), the Innovation Program for Quantum Science and Technology (No. 2021ZD0302502), and Guangdong Major Project of Basic Research, China (Grant No. 2021B0301030003). Y.W. is partially supported by the New Cornerstone Science Foundation through the New Cornerstone Investigator Program and the XPLORER PRIZE. J.-C.Y. acknowledges the financial support from the National Science and Technology Council (NSTC) in Taiwan under grant no. NSTC-112-2112-M-006-020-MY3. This work used the facilities of the National Center for Electron Microscopy in Beijing at Tsinghua University and Nanoport at ThermoFisher Scientific Shanghai.

## Author contributions

Z.C. initiated the idea and designed the studies. Z.D. developed the algorithm, conducted simulations, and analyzed data with the supervision of Z.C. and Y.W. Electron microscopy experiments were performed by Z.D., Y.Z., S.Y.L., and Z.C. Pr-doped $Ca_2Co_2O_5$ films were grown by S.C.L., J. Z., and P.Y. Twisted bilayer STO samples were prepared by C.-C.C., Y.-C.L., and J.-C.Y. Z.D. and Z.C. wrote the manuscript with inputs from all authors.

## Competing interests

The authors declare no competing interests.
