## [Transparent Peer Review file · Nature Communications]

Sub-nanometer depth resolution and single dopant visualization achieved by tilt-coupled multislice electron ptychography

Corresponding Author: Professor Zhen Chen

Version 0:

Reviewer comments:

Reviewer #1

(Remarks to the Author)

The manuscript entitled „Sub-nanometer depth resolution and single dopant visualization achieved by 1 tilt-coupled multislice electron ptychography " reads very interesting and demonstrates ptychographic imaging with very high resolution along the z-direction. However, contrary to their claim that "In the following, we will show that the implementation of sample tilts in electron ptychography opens up new imaging possibilities, in particular the three-dimensional resolving power at a mild illumination, which, however, has never been explored." is completely false in light of the following more than 10 year old references that have been ignored by the authors:

[1] Physical Review B 87 (2013) 18 DOI:10.1103/PhysRevB.87.184108

[2] Comptes Rendus Physique 15 (2014) 119-125 DOI:10.1016/j.crhy.2013.10.004

It seems that this neglectation may not have been accidental, since the corresponding author of the manuscript has coauthored also the following manuscript, in which the first of these two references has been cited:

[3] Optics Express 28 19 28306 DOI:10.1364/OE.396925

Just as the present manuscript, reference [1] includes simulations at limited dose and for small tilt angles that have been approximated using a shifted Fresnel propagator between the slices. Apart from the alignment routine that the authors have had to employ for dealing with experimental data (refs. [1] and [2] only deal with simulated data), the reconstruction principle seems to be very similar or even the same, but definitely not novel. In particular the statement on page 6 "Although TCMEP could be limited by its dependence on a sufficiently small tilt angle for the validity of the interlayer shift approximation, reconstruction with a maximum tilt angle of 10° (~ 0.17 rad, which can still be regarded as a small value) is achievable and reliable, with a depth resolution better than 3 Å, as suggested by the simulations in Extended Data Fig. 9." could have been written in exactly the same manner about refs. [1] or [2], since those also used a tilt angle of up to 10° .

I see the novelty of this manuscript in being the first to apply an established reconstruction principle that has so far only been tested on simulated data to experiments. This is a rather complex experiment, since the same field of view had to be imaged at a few slightly different sample tilts.

Here are a few questions regarding the reconstruction algorithm:

- Although the authors mention that the LSQ-ML algorithm was employed for the reconstructions, they do not explain which hyper-parameters were used, e.g. the Noise model, update-batch size and, most importantly, additional regularization constraints. This makes the paper incomplete, as even the corresponding author of the manuscript highlights the importance of hyper-parameters as a coauthor in Sci Rep 12, 12284 (2022).
- In line 300 the authors claim that the exceptional size of a 4D-STEM dataset may cause the memory overflow. While it may be possible, that the size of the objects transmission function can cause memory overflow, for all iterative ptychographic algorithm one needs only one diffraction pattern at a time. Thus, employing lazy loading can drastically free up the required amount of memory. It would help to be specific how many voxels the object contains, and why they expect memory overflow.

- The authors mention multiple times the effective semi-angle α_{eff} . It would be great if they could provide some theoretical background to link it to the measurement parameters, such as convergence or collection angles. While the authors mention in line 160 and in the description of Figure 1 that the wedge-angle increases linearly, the plot in the upper left part of panel 3K looks rather non-linear. It would thus help to create similar plots for the simulated STO data at various dose conditions to confirm that the linear model is accurate.

Here are a few comments regarding the data analysis performed by the authors:

- While I agree that the phase histogram of Ca columns seems to have a distinct peak, I disagree with the statement on page 5 that "In all Co columns within our reconstruction, the plateau remains consistently flat and well-defined, in stark contrast to the Ca columns where the presence of an additional peak is quite common." The peak in the red curve in Fig. 4c is about 0.05 rad, which is very close to the level of phase noise in the profiles of the Co columns in Fig. 4d. Also the distribution of phases of Co columns (Fig. 4f) is approx. 0.12 rad, about twice as wide as the 'hump' in the red curve. In principle, the authors could have correlated the phase humps with an elemental (EELS or EDXS) map of Pr. I will not ask to do such a demanding experiment, but the least they could do is to correlate the Ca columns that have phase values of about 0.47 rad with columns in which they identify such 'humps' in a statistically meaningful manner. The profiles in Fig. 5f are not at all ideal plateaus with one or two 'humps', but fade out rather slowly towards the surfaces. The authors should not discuss the meaning of an increase in phase, but also what it means when the phase is only half of the 'plateau' value. A proper statistical analysis of the 'Pr-signal' in relation to the general fluctuations of the phase within each of the Ca columns seems to be completely missing.
- There are other potential origins for a bimodal phase distribution of the Ca-columns. One of them would be beam damage. After all, the authors have applied a dose of $9e5e/A^2$.
- In Fig. 4e the authors claim that the distribution of phases for the TCMEP is more narrow and reveals a second hump in contrast to the MEP reconstruction. While the distribution of phase values may indeed be slightly more narrow in the case of TCMEP, the authors seem to have been trying to hide the fact that also the MEP-histogram in the inset shows a bimodal distribution by fitting only a single Gaussian to a data set that may be fitted even better with two Gaussians. There is a clear peak in the distribution at about 0.52 rad also in the inset. On page 5 the authors write "the histogram of Ca columns from the normal MEP reconstruction (inset of Fig. 4e) reveals minimal asymmetric features without statistical significance." This statement is provided without any proof, and I will assume that it is wrong, until the author clearly demonstrates by a detailed statistical analysis that what they write is actually correct.

In light of a) the false claim to of a new reconstruction scheme and b) the poor statistical analysis of the data the paper is not recommended for publication.

Reviewer #2

(Remarks to the Author)

Reviewer #3

(Remarks to the Author)

Dong et al combine multislice ptychography with few-tilt data acquisition to obtain 3D reconstructions of crystalline materials, achieving sub-angstrom lateral resolution and sub-nanometer depth resolution. Notably, the authors reconstruct individual flakes of a twisted strontium titanate heterostructure with sub-nanometer resolution, and demonstrate clear identification of individual Pr dopants in a brownmillerite film. The experimental results are very impressive, important to the community and certainly worthy of publication in Nature Communications. I have several concerns about important experimental details, interpretation of results and reproducibility (particularly with regards to Figures 2 and 3), but I believe these concerns can be addressed with minor revisions. Further details are provided below.

Main comments:

Comment 1: The introductory paragraph omits important references, suggesting that the authors' work was the first to show a relationship between the maximum diffraction angle and depth resolution in diffractive imaging. This is not the case. See the references below:

- Nature Communications volume 3, Article number: 730 (2012)
- Nature volume 463, pages 214–217 (2010)

Comment 2: Is there a z-regularization factor for the TCMEP or MEP data? A list of reconstruction parameters would significantly aid the reproducibility of these results.

Figure 1

Comment 3: It's not clear from the figure whether α_{eff} refers to the collection semi-angle or convergence semi angle

subtended by the probe-forming aperture. Furthermore, the cones of illumination in Figure 1(a) and 1(b) are not clearly distinguished from the cone of the effective semi-angle in Figure 1(b). Furthermore, I believe it would be more helpful if the diffraction patterns were not cropped, to demonstrate the use of dark-field information for extending the depth resolution.

Figure 2

Comment 4: The plot of depth resolution versus dose is very helpful. Why does the depth resolution get slightly worse for larger dose for the low-tilt cases? This suggests that the reconstructions for these data points are not optimized.

Comment 5: Figure 2d – the y axis should emphasize that the phase is normalized. On this point, how do the phase amplitude values change with tilt before normalization?

Comment 6: For the simulations, there is no mention of whether lattice vibrations were accounted for by using frozen phonons to model the potential. This may have a significant effect on the depth resolution and precision achievable. Furthermore, did the authors account for partial spatial and temporal coherence? This is not necessarily crucial as MEP can account for these, but I believe it is worth mentioning in a revised manuscript.

Figure 3

Comment 7: The detailed analysis of experimental depth resolution is generally well-presented and impressive. However, it is not convincing that a linear fit in Figure 3k is the best method to calculate the effective semi angle. I would anticipate a quadratic relationship (which is actually visible for one of the curves) between k_r and k_z , in a similar fashion as reported in [1-3]. Can the authors comment on this? Furthermore, with this in mind, does the effective semi-angle account for the curvature of the Ewald sphere? Some important references are below:

- Optics Express Vol. 24, Issue 25, pp. 29089-29108 (2016)
- Nature volume 463, pages 214–217 (2010)
- Journal of the Optical Society of America A, Vol. 29, Issue 8, pp. 1606-1614 (2012)

Comment 8: I was initially confused by the use of α_{eff} and $\alpha_{\text{eff}} + \theta$. In the current definition, $\alpha_{\text{eff_TCMEP}} = \alpha_{\text{eff_MEP}} + \theta$, but there is no use of the MEP or TCMEP subscript. The authors should add a subscript or at least refer to α_{eff} , when referring to the effective semi-angle of TCMEP reconstructions.

Comment 9: I believe Fourier transforms are needed to demonstrate the presence/absence of individual flakes. The current version of the figure leaves much to interpretation. For example, at a depth of 3.2 nm, I would argue that the Moiré pattern is more visible for the TCMEP reconstruction versus the MEP reconstruction. Is this because MEP has a better depth resolution at the top surface of the sample, or is it because the z-position of the reconstructions are not perfectly aligned for MEP and TCMEP? Fourier transforms for different slices in the main or supplementary figures would help clarify this point.

Comment 10: What is the angular sampling and maximum diffraction angle in the experimental data, and how does this compare to the effective semi-angle? The angular sampling is provided for the simulated data, but there is little discussion about the maximum diffraction angle.

Comment 11: Why is a resolution value of 1.33σ chosen instead of the FW80M ($\sim 1.28\sigma$) reported in previous works? Currently, there are inconsistencies across the field of MEP for resolution measurements, so it would be valuable to readers if the authors explained why they used this definition.

Comment 12: How exactly are the error functions determined from the curve fitting? I think you need to specify 'the residuals of the curve fitting' here (if this is what was actually done).

Figures 4 and 5:

Comment 13: How can you decouple lattice distortions from scan position errors? I think the authors solve this problem via registration of the scan positions for each tilted data set, but this should be clarified to aid the reader

Comment 14: Lines 127-129: "On the low dose side with comparable tilt angles, the improvement of TCMEP is more significant, reaching a depth resolution of around 0.55 nm, which is more than 4 times better than that achieved with MEP". This is an impressive result. It seems related to overlapping bright-field disk data between adjacent tilts, as low-dose single tilt data would have little dark field signal above the noise level. Can you comment on the dose efficiency of multiple bright-field tilts versus fewer dark field tilts? No further data/processing is necessary, but a commentary would be appreciated by readers.

Other

Comment 15: The dashed rectangles in Figure 5(c) are difficult to see and should have thicker lines in a revised manuscript

Comment 16: Line 285: Extended Fig 7A – this should be Extended Fig 6A

Comment 17: Ref. 18 – A quick Google search found a published result for this work: Phys. Rev. Applied 22, 014016 (2024)

Comment 18: Line 66-67: At a mild illumination intensity

Comment 19: Line 223-224: "Despite a slight sacrifice in depth resolution, the results demonstrate much better overall quality and precision." – the authors should cite the paper which demonstrates poorer precision than the results shown in this manuscript. Furthermore, "better overall quality" is quite vague. Please replace with a quantitative comparison and a relevant citation to the literature.

Comment 20: A spelling and grammar check should be performed before submitting a revised manuscript version.

Reviewer #4

(Remarks to the Author)

As a atom sized electron probe transits a specimen it picks up a rich variety about the atomic structure of the target. It has long been recognized within the electron microscopy community that this information is encoded within the electron probe, whilst detailed, is difficult to reconstruct due to the strong multiple scattering of the illumination. There has been recent progress in answering this research question due to the advent of 4D-STEM, where modern cameras record the full 2D diffraction pattern from a focused electron at a speed commensurate with the 1-100 microsecond timescales of the 2D probe scan, and newer reconstruction algorithms. Multislice electron ptychography has thus far proven the most successful algorithm for turning 4D-STEM data into reconstructions of the sample, with demonstrable improvements in resolution once multiple scattering artifacts are successfully removed from data. While resolution improvements are an impressive demonstration of the power of these new algorithms the ability to reconstruct more 3D information (which is usually lost in a typical transmission electron microscope experiment) about the sample is a far more illustrious goal and one that seems to have been achieved with this publication. Whilst 3D information is often reconstructed in electron microscopy using a tomography tilt series these are challenging for slab-like crystalline specimens since electron multiple scattering introduces a strong, non-linear effect on image contrast as a function of tilt. In this work a small tilt (2-4 degrees) is shown to be sufficient to vastly improve the robustness of multislice ptychography such that single atom dopant detection and the resolution of separate layers in a heterostructure are now much better detected. Overall the paper is well written and structured, the demonstrations of the technique on simulated and experimental datasets are well presented and the results are impressive and have the potential to have a large impact on the TEM field going forward. I recommend publication subject to a few minor changes.

Introduction

"Real-space imaging of three-dimensional atomic structures is a critical yet challenging task in materials science" would be more logically worded as "Real-space, three-dimensional imaging of atomic structures in materials science is a critical yet challenging task"

"This technique requires only a moderate level of data acquisition or processing, and can be seamlessly integrated into electron microscopes equipped with conventional components." -> This statement does not strike me as factual, 30 hours of processing on an A100 GPU is hardly moderate and the technique requires at least a high end microscope equipped with a FEG, probe aberration corrector and hybrid pixel detector some of which are not considered "conventional components" yet. Not state of the art specialist equipment but not what will be found at many TEM centers. The statement: "The technique can be applied on a high-end, commercially available TEM with a probe aberration corrector and hybrid pixel detector and data processing is accessible to modern high-performance computing systems." is more accurate.

Principal and reconstruction process

The theoretical explanation of using 3D linear contrast transfer functions to my mind lacks insight. The improvement in depth resolution cannot be explained with linear imaging theory and this is readily seen by a back of the envelope calculation. To a first approximation sufficient for discussion here, a single image in STEM gives information in Fourier space in wedge equal to the probe forming semi-angle. For the 25 mrad semi-angle used in this work this corresponds to 1.4 degrees, so the missing wedge would be $180 - 2 \times 1.4 = 177.2$ degrees! If the sample is tilted by 4 degrees this would missing wedge would reduce to a still pitiful 169.6 degrees which would still be highly unsatisfactory. The actual results in this paper suggest a far more impressive improvement with small tilts than the theoretical analysis (which should give an upper bound in improvements) would suggest and to my mind this is better explained through better diversity of inputs into the phase retrieval algorithm. A hallmark of strong multiple electron scattering by crystalline samples is the strong dependence of TEM results on even small (few mrad) tilts of the specimen so different tilts will give a much more varied input to the (famously unstable without strong regularization see Schloz, Marcel, et al. "Overcoming information reduced data and experimentally uncertain parameters in ptychography with regularized optimization." *Optics Express* 28.19 (2020): 28306-28323.) multislice ptychography algorithm. This improved and more diverse input should result in a much more reliable and more accurate reconstruction of the object function which is what the authors observe. This hypothesis might be tested by comparing ptychography results with and without the small tilt for an object that doesn't give strong multiple scattering, eg. a very small (few nm) crystal nanoparticle, to see if a similarly impressive improvements in depth resolution are realized.

In the TCMEP reconstruction algorithm are the different planes of the object truly shifted in real-space or does the algorithm incorporate a tilted specimen by a tilted Fresnel propagation function (ie. shifted in reciprocal space) as a standard multislice calculation would account for tilts? The latter strikes me as more accurate though the former might be serviceable.

Perhaps state clearly that the tilts used a specimen (ie. stage) tilts, beam tilts are an alternative though this will likely induce

aberrations which must then be solved jointly in the reconstruction.

Simulation on imaging with a single dopant

Could the authors please include an image of the depth section of the dopant atom for the different maximum tilts in Fig. 2? Even for high quality electron tomography work these give an unflattering image of the quality of reconstruction but can reveal the extent of the missing wedge.

Depth resolution to sub-nanometers

Could the authors include inset images of the Fourier transform of each reconstructed slice in Fig 3? This will give a better visual indication of layer resolution. The atomic model from vesta in 3 (i) could also be tweaked by deleting the atoms from each layer of the moire stack for some fraction of the supercell so that the individual SrTiO₃ lattices can be seen and how they form a Moire in projection.

I disagree with the analysis presented in Fig. 3 (k), the fitted straight lines (dashed lines) do not match the experimental measured profiles of k_r vs k_z well at all since the straight line fit is constrained to start at the origin so I don't believe the estimates of missing wedge that flow from this analysis. I think this part should be removed from the paper. The same applies to Fig. 5c of the supplementary.

Alignment of different tilts and the reconstruction process

This step seems critical to the whole algorithm working well and I'm impressed that the approach outlined in this section has worked as well as it has.

For one of the experiments, eg. Fig 3, could the authors please present as a supplementary figure showing the alignment of the separate ptychographic reconstructions for each acquisition in the tilt series and a plot showing the change in refined probe positions from the drift-correction algorithm built into the ptychographic reconstruction?

These citations should be added to the manuscript due to the strong overlap of the work.

Schloz, Marcel, et al. "Improved Three-Dimensional Reconstructions in Electron Ptychography through Defocus Series Measurements." arXiv preprint arXiv:2406.01141 (2024).

Ren, David, et al. "A multiple scattering algorithm for three dimensional phase contrast atomic electron tomography." Ultramicroscopy 208 (2020): 112860.

Version 1:

Reviewer comments:

Reviewer #1

(Remarks to the Author)

The authors have responded satisfactorily to many of the detailed remarks in the previous comment. However, the main criticism was that they have claimed to having developed a novel reconstruction principle (TCMEP) and in their rebuttal they write: "we would like to highlight that the artificial neural network (ANN) framework used in Refs. [1,2] differs from the tilt-coupled multislice electron ptychography (TCMEP) utilized in our study."

I ask the authors to explicitly state the difference. While the new references [25,26] in the manuscript mention the analogy to ANN in their manuscript before describing explicitly the analytical expressions for the gradients they use in their CG optimization, reference [54] cited by the authors for the LSQML principle applied by them provides expressions for the gradients without referring to ANNs and also apply CG optimization. While ref. [25] uses a loss-function in form of the sum of squared differences, ref. [57], which the corresponding author is co-author on, compares different loss functions, including the log-likelihood that is also the basis for the LSQML optimization used in this manuscript. So, I would say that in terms of the reconstruction principle, the authors have reimplemented and applied already published algorithms. This should be made very clear in the manuscript. In their revised version, the authors write already in the abstract "Here, we introduce a new algorithm based on multislice electron ptychography, ..." – In light of the above argument, this is very deceiving. Also other places, they write "Our approach..." – Apart from the alignment of probe positions across data sets of different tilt (see below), others have developed this "approach" first.

The authors continue to write in their rebuttal: "Several critical experimental factors—such as the retrieval of the incident probe function, correction of probe position errors, and alignment across different datasets—were not addressed in Refs. [1,2], yet these are essential prerequisites for successful experimental reconstructions." I again have to object. The update of probe position and probe wave function is standard in ptychography. Ref. [1] has a separate section in the appendix with the explicit expression for the gradient with respect to the defocus of the probe. Ref. [57] (same authors as [1,2]) devotes also a detailed analysis to the refinement of the probe positions and full probe wave function (see e.g. Figs. 3 & 4 in ref. [57] and all of section 3). Also the alignment across different data sets of different defocus has been done by the same authors of in ref. [49].

Here is one more example (exemplary for other places in ms) of where I disagree with the authors interpretation of ref. [25]. They write: "Simulations show that the most significant improvement occur at small tilt angles, with atomic-scale depth resolution achievable at tilt angles of approximately 4°, corroborating previous proof-of-principle demonstrations using similar reconstruction schemes [25,26]."

I agree with the authors (and have stated this in my report on the first version) that this is the first experimental implementation of including multiple tilts in the reconstruction. However, apart from the refinement of tilt angles for each dataset and the alignment of probe positions between data sets of different tilt, I do not at all agree with the argument of novelty by the authors.

(Remarks on code availability)

Reviewer #2

(Remarks to the Author)

(Remarks on code availability)

Reviewer #3

(Remarks to the Author)

I want to thank Dong et al. for comprehensively addressing all of my comments from the previous review. The paper reads well and explains the research such that it can be reproduced by others. I recommend this article for publication. I have a few minor comments that could be addressed to improve the article quality:

- The use of the words 'remarkable' and 'unprecedented' are not valid here, as I believe existing theory would suggest that this experiment could be achieved. I think the authors should focus on quantitative values to demonstrate the significance of their result. Similarly, the use of 'only moderate resolutions' for other people's work is unfair and subjective. This should be replaced with something more objective, i.e. the ratio of depth resolution vs lateral resolution.

- line 58: proof-of-principle is spelled incorrectly
- line 185: values of lateral resolution would be welcome here

(Remarks on code availability)

There was no code provided.

Reviewer #4

(Remarks to the Author)

To my reading this is a sound piece of research and the authors have comprehensively addressed some of the suggestions I have made in my previous review. I recommend publication of the work.

(Remarks on code availability)

The code is listed as "available upon reasonable request", best practice is to upload the code to Github and to create a release on Zenodo

We appreciate the insightful comments provided by the reviewers. In response to the comments of the referees, we have made substantial revisions to the manuscript. The revisions of the main text have been marked in red. The referees' comments are in black and the authors' responses in blue.

Response to Referee: 1

The manuscript entitled "Sub-nanometer depth resolution and single dopant visualization achieved by 1 tilt-coupled multislice electron ptychography" reads very interesting and demonstrates ptychographic imaging with very high resolution along the z-direction. However, contrary to their claim that "In the following, we will show that the implementation of sample tilts in electron ptychography opens up new imaging possibilities, in particular the three-dimensional resolving power at a mild illumination, which, however, has never been explored." is completely false in light of the following more than 10 year old references that have been ignored by the authors:

[1] *Physical Review B* 87 (2013) 18 DOI:10.1103/PhysRevB.87.184108

[2] *Comptes Rendus Physique* 15 (2014) 119-125 DOI:10.1016/j.crhy.2013.10.004

It seems that this neglection may not have been accidental, since the corresponding author of the manuscript has coauthored also the following manuscript, in which the first of these two references has been cited:

[3] *Optics Express* 28 19 28306 DOI:10.1364/OE.396925

Just as the present manuscript, reference [1] includes simulations at limited dose and for small tilt angles that have been approximated using a shifted Fresnel propagator between the slices. Apart from the alignment routine that the authors have had to employ for dealing with experimental data (refs. [1] and [2] only deal with simulated data), the reconstruction principle seems to be very similar or even the same, but definitely not novel. In particular the statement on page 6 "Although TCMEP could be limited by its dependence on a sufficiently small tilt angle for the validity of the interlayer shift approximation, reconstruction with a maximum tilt angle of 10° (~ 0.17 rad, which can still be regarded as a small value) is achievable and reliable, with a depth resolution better than 3 \AA , as suggested by the simulations in Supplementary Fig. 9." could have been written in exactly the same manner about refs. [1] or [2], since those also used a tilt angle of up to 10° .

I see the novelty of this manuscript in being the first to apply an established reconstruction principle that has so far only been tested on simulated data to experiments. This is a rather complex experiment, since the same field of view had to be imaged at a few slightly different sample tilts.

We are grateful for the referee's description of our work as "very interesting" and appreciate the mention of previous studies that demonstrated ptychography with a similar tilt-coupled strategy using simulated data, as noted in Refs. [1,2]. These prior works certainly deserve acknowledgments, and we sincerely apologize for the unintentional oversight of these publications. There also appears to be a misunderstanding regarding the recognition of tilt-coupled strategy in Refs. [1,2]. Although the corresponding author of this manuscript is indeed

a coauthor of Ref. [3], he is not the lead author, which may lead to some details being overlooked. Nevertheless, we take full responsibility for the omission and have revised the manuscript to properly acknowledge the mentioned works. While acknowledging the referee's points, we would like to highlight that the artificial neural network (ANN) framework used in Refs. [1,2] differs from the tilt-coupled multislice electron ptychography (TCMEP) utilized in our study. TCMEP employs a reconstruction framework based on the least-squares maximum-likelihood (LSQ-ML) optimization algorithm, whereas the reconstructions in Refs. [1,2] were ANN-based. Additionally, the coincidence of a maximum tilt angle of 10° in both our manuscript and in Refs. [1,2] is not surprising, as John Cowley's classical textbook states that '*tilted beam (or translated object) approximations are suitable only for tilts of a few degrees*' (Cowley J.M., *Diffraction Physics*, Ch. 11, now added as a reference).

Several critical experimental factors—such as the retrieval of the incident probe function, correction of probe position errors, and alignment across different datasets—were not addressed in Refs. [1,2], yet these are essential prerequisites for successful experimental reconstructions. As the referee also acknowledged, the novelty of our manuscript lies in the *first* experimental realization of this *rather complex technique with very high resolution along the z-direction*. It is widely accepted that the experimental implementation of phase retrieval algorithms demands significantly more effort than the initial proof-of-principle demonstrations. For example, the robust experimental realization of multislice electron ptychography by Z. Chen, D. Muller, *et al.*, in 2021 (*Science* 372, 826 (2021)) sparked renewed interest and applications in electron ptychography, despite the basic algorithm being introduced nearly a decade earlier in 2012 (*J. Opt. Soc. Am. A* 29, 1606 (2012)).

We would like to emphasize the significant efforts we invested in alignment and convergence during the experimental implementation of TCMEP. Successful reconstruction from experimental datasets required the precise alignment of 4D-STEM data prior to TCMEP reconstructions (Fig. R1), refinement of tilt angles for each dataset (Fig. R2b), and accurate correction for sample drift (Fig. R2c). These critical steps, which are essential for the realization of our experimental approach, are now detailed in the revised Supplementary Figs. 12-13. We believe this experimental advancement will find wide applications in solving complex three-dimensional structures, such as point defects and related structural distortions, as demonstrated in our manuscript.

Fig. R1| 4D-STEM data alignment prior to TCMEP reconstruction. **a-c**, MEP reconstruction results for the full datasets acquired at tilt angles of 0° (**a**), -1° (**b**), and $+1^\circ$ (**c**). The intentional electron-beam-induced defect area is used as a reference for alignment, with the region of interest (ROI) highlighted by a blue square. **d-f**, MEP reconstruction results for the cropped ROI within the datasets acquired at tilt angles of 0° (**d**), -1° , (**e**) and $+1^\circ$ (**f**). The results demonstrate good alignment and are used to initialize the TCMEP reconstruction, while the slight residual misalignments will be further refined during the iterations.

Fig. R2| Details of the TCMEP reconstruction for experimental datasets on twisted SrTiO₃. **a**, The initial and reconstructed mixed-state probes for each dataset in TCMEP. **b**, The refinement of tilt angles for each dataset. **c**, The refined probe-positions for each dataset.

In response to the referee's suggestions, we have revised the relevant sections of the manuscript, moderated our claims, and included the appropriate citations.

Changes made:

- In the 2nd paragraph in *Introduction* section, we include citations to the suggested references.

Simulations show that the most significant improvement occur at small tilt angles, with atomic-scale depth resolution achievable at tilt angles of approximately 4°, **corroborating previous proof-of-principle demonstrations using similar reconstruction schemes**^{25,26}. Our experiments with TCMEP achieve sub-nanometer depth resolution, successfully transferring higher-frequency information along the depth dimension through the sample tilt series.

- In the 1st paragraph in *Principle and reconstruction process* section, we temper our claims and include the suggested references.

In the following sections, we will demonstrate how specimen tilts in electron ptychography unlock new imaging possibilities, especially by enhancing 3D resolution **at relatively mild illumination intensities, which has only been explored in proof-of-principle studies using simulated datasets**^{25,26}.

- In the 2nd paragraph in *Discussion* section, we include the suggested references.

Although TCMEP's reliance on a small tilt angle for the interlayer shift approximation might seem a limitation, we show that reconstructions using a maximum tilt angle of 10° (~0.17 rad, still within the small-angle regime⁴²) are both feasible and reliable. This approach achieves a depth resolution of better than 3 Å (Supplementary Fig. 10), **which agrees well with previous simulation results using a comparable approach**^{25,26}.

- We have included a citation to 'Schloz, M. *et al.*, *Opt. Express*, **28**, 28306 (2020)' in the Methods section.
- Figures R1 and R2 are provided as new Supplementary Figures 12 and 13.

Hereafter, we will address the comments from the referee point-by-point as follows:

Here are a few questions regarding the reconstruction algorithm:

• *Although the authors mention that the LSQ-ML algorithm was employed for the reconstructions, they do not explain which hyper-parameters were used, e.g. the Noise model, update-batch size and, most importantly, additional regularization constraints. This makes the paper incomplete, as even the corresponding author of the manuscript highlights the importance of hyper-parameters as a coauthor in Sci Rep 12, 12284 (2022).*

We appreciate the referee's suggestion to highlight the missing reconstruction parameters in our paper. A comprehensive table of these parameters is now provided in Supplementary Table 1. Additionally, we would like to clarify that the optimization of hyperparameters using methods such as Bayesian optimization, as discussed in Sci Rep 12, 12284 (2022), is only necessary when the imaging parameters are not well-optimized or when operating under low-dose conditions. In our case, the convergence of the experimental parameters is very good, as shown in Fig. R2. To avoid any confusion, we have added this clarification to the Methods section.

Changes made:

- A complete table of parameters is provided as Supplementary Table 1.
- We also clarify that Bayesian optimization is not always necessary in TCMEP reconstructions, as revised in the *Methods-Alignment of datasets and reconstruction process* section:

Bayesian optimization for hyperparameter refinement⁵⁶ was not applied in this study, as conventional TCMEP reconstruction parameters, listed in Supplementary Table 1, led to sufficiently high convergence, consistent with previous report for MEP¹⁶.

• *In line 300 the authors claim that the exceptional size of a 4D-STEM dataset may cause the memory overflow. While it may be possible, that the size of the objects transmission function can cause memory overflow, for all iterative ptychographic algorithm one needs only one diffraction pattern at a time. Thus, employing lazy loading can drastically free up the required amount of memory. It would help to be specific how many voxels the object contains, and why they expect memory overflow.*

We appreciate the referee's valuable questions. The object voxels are specified in Supplementary Table 1. Our codes are optimized for GPU memory requirements, with most of the memory allocated for storing intermediate exit wave functions for each slice, which are used to estimate the update direction during back-propagation. For example, the required GPU memory for reconstructing twisted SrTiO₃ is estimated as follows: Total GPU memory = Number of datasets (4) × Batch size (100) × Number of layers (40) × Size of complex exit wave functions after each layer (124×124×2) × Number of probe modes (4) × FP32 size (4 Bytes) ≈ 10 GB. Reconstruction for 1000 iterations using these parameters on an A100 GPU takes approximately 30 hours. A slight change in parameters (e.g., using 8 probe modes) may result in a memory overflow on an RTX 4090 (with 24 GB memory). One can, of course, use lazy loading in MEP/TCMEP reconstructions (such as reducing batch size or processing diffraction patterns sequentially) to lower the GPU memory requirement, but this will significantly increase the iteration time. For instance, reducing the batch size to 25 would free up 75% of the GPU memory, but the total reconstruction time would extend to ~60 hours. The parameters should be selected to balance memory requirements with total reconstruction time. Of course, these requirements are based on our current reconstruction scheme and there should be other optimized ways to circumvent these problems. Therefore, we have revised the manuscript to include these important discussions.

Changes made:

- The discussion about memory requirement in *Methods* section is revised to improve clarity:

The parameters in Supplementary Table 1 were carefully selected to balance memory requirements with total reconstruction time based on our current implementation. For example, TCMEP reconstruction for twisted SrTiO₃ with a maximum tilt of 2° required approximately 10 GB of GPU memory and 30 hours of computation time. Therefore, advancements in both hardware capabilities and algorithmic efficiency are crucial to accelerate the computational process for larger datasets, especially when striving for the ultimate atomic-scale depth resolution.

• *The authors mention multiple times the effective semi-angle α_{eff} . It would be great if they could provide some theoretical background to link it to the measurement parameters, such as convergence or collection angles. While the authors mention in line 160 and in the*

description of Figure 1 that the wedge-angle increases linearly, the plot in the upper left part of panel 3K looks rather non-linear. It would thus help to create similar plots for the simulated STO data at various dose conditions to confirm that the linear model is accurate.

We are grateful to the referee for these thoughtful suggestions. First, we have replaced the term α_{eff} with $\beta_{\text{MEP/TCMEP}}$ in the main text for improved clarity. The effective semi-angle ($\beta_{\text{MEP/TCMEP}}$) for 3D information transfer is influenced by both the maximum diffraction angle (α_{max}) and the convergence angle (α_{conv}), as described in a recent preprint (arXiv:2407.18063). While α_{max} sets the ultimate limit for depth information in strong scattering, most of the 3D information transfer falls within the weak-scattering regime, which is also governed by α_{conv} . Therefore, under finite—and especially low—dose conditions, the probe convergence angle significantly influences the practical limits of information transfer.

We have also generated the k_z versus k_r plots for the simulated datasets, as shown in Fig. R3a with details in Fig. R4, revealing a quadratic relationship for 3D information transfer, which aligns with the findings in arXiv:2407.18063. In Fig. R3a, the effective semi-angles $\beta_{\text{MEP/TCMEP}}$, defined by the slopes of fitted curves near the origin, are measured at 207 mrad (11.8°), 352 mrad (20.1°), and 553 mrad (31.6°), respectively. We agree with the referee that the k_z - k_r relationship shows a nonlinear dependence across the full frequency range. The original linear fit was only intended to highlight the opening angle at low lateral frequencies, which may be dependent on the chosen threshold of the frequency range. Therefore, we have revised the manuscript to clarify this and avoid such confusion. Since the wedge angle increments are significantly larger than the tilt angle increments, we have removed the equation $\beta_{\text{TCMEP}} = \beta_{\text{MEP}} + \theta$ and the linear fit from the original Fig. 3. Instead, we incorporated Fig. R3b into Fig. 3 to emphasize the improvement in depth resolution across all lateral spatial frequencies in experimental results.

It is also important to address the extracted $\beta_{\text{MEP/TCMEP}}$ values. In arXiv:2407.18063, the β_{MEP} extracted for experimental datasets is ~ 70 mrad, while in our simulations, β_{MEP} is ~ 207 mrad—nearly three times larger. This difference primarily arises from the idealized simulation conditions, which do not account for experimental uncertainties such as sample drift or partial spatial-temporal coherence. The relatively large β_{MEP} value observed in simulations highlights the capability of MEP techniques to resolve depth information, particularly under improved imaging conditions.

Fig. R3| Depth resolutions and k_z as a function of lateral spatial frequencies in TCMEP. **a**, Information transfer boundary for simulated datasets as a function of lateral spatial frequency, fitted with a quadratic function passing through the origin. **b**, Depth resolution as a function of lateral spatial frequencies for experimental datasets of twisted bilayer SrTiO₃.

Fig. R4| Depth resolution of TCMEP versus spatial frequency in simulations. **a**, Typical FFT image of a reconstructed SrTiO₃ crystal in simulations. **b**, Depth resolution for each Bragg peak under different tilt conditions, extracted following the procedure in panels **c-e**. **c-e**, Depth profiles of five lateral spatial frequencies indicated by circles in panel **a**, corresponding to maximum tilts of 0° (**c**), 2° (**d**), and 4° (**e**), respectively. Gaussian error functions are used to fit the data points and determine depth resolutions at each lateral frequency. Error bars are derived from residuals of curve fitting.

Changes made:

- A discussion about the relationship between $\beta_{\text{MEP/TCMEP}}$, α_{conv} , and α_{max} is provided in the *Principle and reconstruction process* section:

To make a direct comparison with conventional focal-series ADF imaging, the 3D information transfer via MEP from a single dataset is qualitatively modeled as a cone with an effective semi-angle, β_{MEP} . This angle generally depends on the maximum diffraction angle, but is also

constrained by the probe's convergence angle under finite—and particularly low—dose conditions^{16,19,28,29}.

- We no longer adopt the relation $\beta_{TCMEP}=\beta_{MEP}+\theta$ in Fig.1b.
- Fig. R3a is added to main Fig. 2. We also provided corresponding discussions in the main text:

The Fourier transform of the reconstructed phase image reveals the boundary of information transfer¹⁹ (Fig. 2h, details in Supplementary Fig. 2), which qualitatively agrees with the schematic illustration in Fig. 1b, showing that the boundary expands along the depth dimension across all lateral spatial frequencies. The effective semi-angles $\beta_{MEP/TCMEP}$, defined by the slopes of fitted quadratic curves near the origin, are measured at 207 mrad (11.8°), 352 mrad (20.1°), and 553 mrad (31.6°), respectively. Notably, β_{MEP} in our simulation is around three times larger than the corresponding value from experimental results reported in ref.¹⁹. This discrepancy primarily arises from the idealized simulation conditions, which do not account for experimental imperfection such as sample drift or partial spatial-temporal coherence. Moreover, the unavoidable roughness of sample surface broadens the depth distribution in the Fourier spectra, leading to an overestimation of depth resolution in the experimental results. Nevertheless, it has been confirmed that the improvement in depth resolution is primarily driven by information gathered at higher angles.

- Fig. R3b is added to Fig. 3. We also provided corresponding discussions in the main text: Moreover, the corresponding Fourier analysis (Fig. 3e, details in Supplementary Fig. 6) reveals a universally improved depth resolution across all lateral spatial frequencies as the maximum tilt angle increases, consistent with the findings in Fig. 2h for the simulated datasets.

- Fig. R4 is provided as Supplementary Fig. 2.

Here are a few comments regarding the data analysis performed by the authors:

• While I agree that the phase histogram of Ca columns seems to have a distinct peak, I disagree with the statement on page 5 that "In all Co columns within our reconstruction, the plateau remains consistently flat and well-defined, in stark contrast to the Ca columns where the presence of an additional peak is quite common." The peak in the red curve in Fig. 4c is about 0.05 rad, which is very close to the level of phase noise in the profiles of the Co columns in Fig. 4d. Also the distribution of phases of Co columns (Fig. 4f) is approx. 0.12 rad, about twice as wide as the 'hump' in the red curve. In principle, the authors could have correlated the phase humps with an elemental (EELS or EDXS) map of Pr. I will not ask to do such a demanding experiment, but the least they could do is to correlate the Ca columns that have phase values of about 0.47 rad with columns in which they identify such 'humps' in a statistically meaningful manner. The profiles in Fig. 5f are not at all ideal plateaus with one or two 'humps', but fade out rather slowly towards the surfaces. The authors should not only discuss the meaning of an increase in phase, but also what it means when the phase is only half of the 'plateau' value. A proper statistical analysis of the 'Pr-signal' in relation to the general fluctuations of the phase within each of the Ca columns seems to be completely missing.

We greatly appreciate the referee's insightful questions and suggestions. First, we acknowledge that comparing Co and Ca columns is not straightforward due to their differing atomic numbers ($Z=27$ versus $Z=20$). In line with the referee's recommendation, we have provided additional microscopic information regarding Ca columns with and without Pr dopants. To this end, we performed additional elemental EELS mapping of Pr M_5 -edge (Fig. R5b), which clearly shows that Pr preferentially substitutes into the Ca2 sites (indicated by red arrows), rather than the Ca1 sites (blue arrows). This substitution spontaneously breaks the lattice inversion symmetry, and the accurate imaging of Pr signal with MEP/TCMEP should distinguish between the inequivalent Ca sites, separating it from intrinsic phase fluctuations.

This distinct feature is captured in the depth-sectional images in Fig. R5d (for MEP) and Fig. R5f (for TCMEP). We have also provided a complete set of depth-sectional images for all Ca1 and Ca2 columns in Figs. R6 and R7, respectively. At this point, we are confident in presenting the statistical histogram of phase values for the Ca and Co columns (Fig. R8). The phase distribution for Ca1 is symmetrically centered around ~ 0.4 rad, while the Ca2 phase distribution reveals an additional shoulder at ~ 0.5 rad in the TCMEP results (Fig. R8c). These findings, in agreement with the EELS mapping, confirm that the additional phase peaks correspond to substitutional Pr atoms.

We would also like to address the *seemingly* broader distribution of Co phases. In Fig. R8, we have included the mean values and standard deviations for all atoms, which demonstrate a consistent uncertainty of $\sim 8\%$ in phase values (8% for Ca1 and 9% for Co). The uncertainties in Ca and Co phases are of the same order, and the broader distribution in Co phases is simply a result of its larger mean values. Additionally, the phase enhancement from Pr dopants ($\sim 20\%$) is significantly above the intrinsic uncertainty (8%), further validating our approach in identifying Pr dopants. It is also worth noting that all phase values from TCMEP are systematically smaller than those from MEP by ~ 0.03 rad, which can be attributed to a global phase difference between the reconstructions. This is common in MEP reconstructions, as the absolute phase value is physically meaningless unless calibrated using a real vacuum region, which is absent in our datasets.

Fig. R5| Experimental imaging of Pr dopants in $(\text{Pr}_{0.05}\text{Ca}_{0.95})_2\text{Co}_2\text{O}_5$ thin film. **a**, Crystal structure of the $(\text{Pr}_{0.05}\text{Ca}_{0.95})\text{Co}_2\text{O}_5$ film grown on a LaAlO_3 substrate. Two distinct calcium atom rows are marked with blue and red arrows, corresponding to Ca1 and Ca2, respectively. **b**, Simultaneously acquired STEM-HAADF image (top) and Pr M_5 -edge intensity map from STEM-EELS (bottom). The Ca2 rows are indicated by red arrows based on the HAADF image. **c**, Projected phase image reconstructed using MEP without tilting the sample. **d**, Depth profiles corresponding to panel **c** for Ca1 (left) and Ca2 (right). **e**, Projected phase image of the same region reconstructed using TCMEP with a maximum tilt of 1° . **f**, Depth profiles corresponding to panel **e** for Ca1 (left) and Ca2 (right), respectively. The Ca2 column, marked by a vertical black arrow, shows a 4 nm difference in depth between two nearby peaks, highlighting the enhanced depth resolution achieved in TCMEP reconstructions. Scale bars, 1 nm.

Fig. R6| Depth sectioning images for all Ca1 columns in the TCMEP reconstruction. **a**, Crystal structure of the $(\text{Pr}_{0.05}\text{Ca}_{0.95})\text{Co}_2\text{O}_5$ film (left) and projected phase image reconstructed using TCMEP(right). **b-f**, Depth sectioning images corresponding to the five arrows along Ca1 rows denoted by numbers 1 to 5 in panel **a**.

Fig. R7| Depth sectioning images for all Ca2 columns in the TCMEP reconstruction. a, Crystal structure of the $(\text{Pr}_{0.05}\text{Ca}_{0.95})\text{Co}_2\text{O}_5$ film (left) and projected phase image reconstructed using TCMEP(right). **b-f,** Depth sectioning images corresponding to the five arrows along Ca2 rows denoted by numbers 1 to 5 in panel a.

Fig. R8| Statistics of phase values for Ca1, Ca2 and Co columns in MEP and TCMEP reconstructions. a-b, Statistical results of phase values for Ca (a) and Co (b) sites in MEP results. **c-d,** Statistical results of phase values for Ca (c) and Co (d) sites in TCMEP results.

Changes made:

- Figure 4 is revised; Figures R6-R8 are included in Supplementary Figures 7, 8 and 11.
- We added a description about the Pr-dopant-induced inversion symmetry breaking in the sample and how this can be utilized to exclude potential origins of phase peaks other than Pr dopants:

The unique advantage of this material lies in the spontaneous breaking of lattice inversion symmetry after Pr doping, as evidenced by the electron energy-loss spectroscopic (EELS) mapping in Fig. 4b-c. Pr substitutes preferentially occupy the Ca2 sites (indicated by red arrows) over the Ca1 sites (blue arrows), despite the equivalence of these sites in the undoped parent phase. This selective substitution could be related to spontaneous polar distortions observed in similar compounds³⁶. Consequently, precise imaging of Pr dopants via MEP or TCMEP must distinguish these inequivalent Ca sites, allowing for the differentiation of real atomic substitutions from intrinsic phase fluctuations, imaging artifacts or beam damage effects.

We use only three sample tilts at 0° and $\pm 1^\circ$ to reduce the experimental workload. Figures 4d and 4g show the projected phase images reconstructed with MEP and TCMEP, both achieving similar lateral resolution. To identify Pr substitution in the Ca columns, depth sectioning was conducted along the Ca1 (blue arrow) and Ca2 (red arrow) rows (Fig. 4e and 4h, depth profiles for all Ca rows are provided in Supplementary Figs. 7-8). A clear distinction is observed between Ca1 and Ca2 in both reconstructions, with Ca1 showing consistently uniform phase distribution, while Ca2 exhibits randomly distributed additional phase peaks. These peaks result from the increase in average atomic number Z when Pr atoms ($Z = 59$) replace Ca atoms ($Z = 20$). As phase value approximately scales with $Z^{0.67}$, the phase associated with Pr atoms is expected to be roughly double that of Ca atoms with the same atomic density. However, due to partial Pr substitution and depth-resolution-induced broadening, the observed phase enhancement from Pr is approximately 20% in our experimental results. Importantly, multiple dopant atoms can be detected within the same atomic column using TCMEP.

• *There are other potential origins for a bimodal phase distribution of the Ca-columns. One of them would be beam damage. After all, the authors have applied a dose of $9e5e/A^2$.*

We greatly appreciate the referee for raising this critical point. As mentioned in our earlier response, the bimodal distribution observed is primarily due to the specific Ca2 columns. The fact that the phase values for Ca1 exhibit a normal distribution rules out beam damage as the cause of the bimodal distribution in the Ca columns, since beam damage would affect both Ca1 and Ca2 columns similarly.

Changes made:

● We exclude the beam damage effect as the possible cause of the phase distribution:

Consequently, precise imaging of Pr dopants via MEP or TCMEP must distinguish these inequivalent Ca sites, allowing for the differentiation of real atomic substitutions from intrinsic phase fluctuations, imaging artifacts or beam damage effects.

• *In Fig. 4e the authors claim that the distribution of phases for the TCMEP is more narrow and reveals a second hump in contrast to the MEP reconstruction. While the distribution of phase values may indeed be slightly more narrow in the case of TCMEP, the authors seem have been trying to hide the fact that also the MEP-histogram in the inset shows a bimodal distribution by fitting only a single Gaussian to a data set that may be fitted even better with two Gaussians. There is a clear peak in the distribution at about 0.52 rad also in the inset. On page 5 the authors write "the histogram of Ca columns from the normal MEP reconstruction (inset of Fig. 4e) reveals minimal asymmetric features without statistical significance." This statement is provided without any proof, and I will assume that it is wrong, until the author clearly demonstrate by a detailed statistical analysis that what they write is actually correct.*

We agree that statistical analysis alone cannot fully demonstrate the performance of TCMEP and MEP. Given our substantial revisions to Figure 4, we have added the depth-sectional images in Fig. R9 to highlight the improvement of TCMEP over MEP.

Both reconstructions show a similar phase distribution for the Ca2 rows, confirming that TCMEP does not introduce extrinsic artifacts or overlook intrinsic features. We emphasize the second Ca2 column, marked by the black arrow, where TCMEP successfully resolves two adjacent peaks approximately 4 nm apart in the depth direction, a feature obscured in the MEP result due to its inferior depth resolution.

As discussed above using Fig. R8, the statistical results for Ca2 phases highlight the improved depth resolution achieved by TCMEP. As the referee noted, both MEP (Fig. R8a) and TCMEP (Fig. R8c) exhibit an asymmetric feature in the Ca2 phase distribution. However, the TCMEP results show a more pronounced and well-defined shoulder centered around 0.5 rad, whereas MEP only reveals an extended tail. The effect of such a continuum phase intensity distribution is more significant and obvious in the real space smearing of Pr dopants as discussed above in Fig. R5. These observations emphasize TCMEP's superior ability to resolve finer structures in the depth dimension.

Fig. R9| Depth sectioning images for four Ca2 columns. **a**, Projected phase image reconstructed using TCMEP. **b-c**, Depth profiles corresponding to Ca2 along the red arrow in panel **a**, for MEP (**b**) and TCMEP (**c**) reconstructions. The vertical black arrow marks a Ca2 column exhibiting a 4 nm difference in depth between two nearby peaks, highlighting the enhanced depth resolution achieved in TCMEP reconstructions. Scale bars, 1 nm.

Changes made:

- We demonstrate the improvement of TCMEP using the depth sectioning curves instead, and the texts regarding statistics are revised to be more precise:

We then compare the depth profiles shown in Fig. 4e and 4h. Both reconstructions show similar phase distributions for the Ca2 rows, confirming that TCMEP does not introduce extrinsic artifacts or obscure intrinsic features. A key difference appears in the second Ca2 column (marked by black arrows), where TCMEP successfully resolves two adjacent peaks separated by approximately 4 nm in depth. This feature is obscured in the MEP reconstruction, due to its inferior depth resolution. Additionally, statistical analysis of phase values for the Ca2 sites reveals an extra shoulder in the TCMEP results (Fig. 4i), which broadens into an extended tail in the MEP reconstructions (Fig. 4f). These findings underscore TCMEP's superior depth resolution, enabling the identification of finer structures along the depth axis.

In addition to the referee's comments, we have also revised Fig. 5 and the corresponding text due to the substantial change in Fig. 4. We illustrate the relationship between Pr dopants and lattice distortions directly through depth profiles and slice images.

Changes made:

- Figure 5, Supplementary Video, and the corresponding main text are revised.

In light of a) the false claim to of a new reconstruction scheme and b) the poor statistical analysis of the data the paper is not recommended for publication.

We have made substantial revisions in response to the referee's comments, and we believe the revised manuscript more precisely and convincingly presents the TCMEP reconstruction scheme as well as the dopant structures in the reconstructions.

Response to Referee: 2

We greatly appreciate the collaborative effort of the referee in reviewing the manuscript and will take into account all the feedback provided.

Response to Referee: 3

Dong et al combine multislice ptychography with few-tilt data acquisition to obtain 3D reconstructions of crystalline materials, achieving sub-angstrom lateral resolution and sub-nanometer depth resolution. Notably, the authors reconstruct individual flakes of a twisted strontium titanate heterostructure with sub-nanometer resolution, and demonstrate clear identification of individual Pr dopants in a brownmillerite film. The experimental results are very impressive, important to the community and certainly worthy of publication in Nature Communications. I have several concerns about important experimental details, interpretation of results and reproducibility (particularly with regards to Figures 2 and 3), but I believe these concerns can be addressed with minor revisions. Further details are provided below.

We greatly appreciate the referee's positive feedback on our work. We have carefully addressed the suggested revisions to improve the clarity and presentation of the manuscript. A point-by-point response to the details of these revisions is provided below.

Main comments:

Comment 1: The introductory paragraph omits important references, suggesting that the authors' work was the first to show a relationship between the maximum diffraction angle and depth resolution in diffractive imaging. This is not the case. See the references below:

- *Nature Communications volume 3, Article number: 730 (2012)*
- *Nature volume 463, pages 214–217 (2010)*

We appreciate the referee for recommending these valuable references. We have revised the second paragraph of the introduction to incorporate the references as Refs. [23,24] and highlighted their impact on our research.

Changes made:

- The 2nd paragraph is revised as follows:

In diffractive imaging techniques like MEP, depth resolution is primarily governed by the maximum scattering angle captured by the detectors^{23,24}. To address these limitations, we introduce tilt-coupled multislice electron ptychography (TCMEP), which employs a moderate probe-forming semi-angle while intentionally tilting the sample off-axis to capture higher-angle scattering information.

Comment 2: Is there a z-regularization factor for the TCMEP or MEP data? A list of reconstruction parameters would significantly aid the reproducibility of these results.

We appreciate the referee's suggestion. In our TCMEP reconstruction for simulated datasets, the z-regularization factor was set at 0.10. For reconstructions of twisted SrTiO₃, regularization factors were 0.30 for MEP and 0.15 for TCMEP. In reconstructions for (Pr, Ca)₂Co₂O₅ thin films, regularization factors were 0.30 for MEP and 0.10 for TCMEP. Smaller

regularization factors were used exclusively for TCMEP, as MEP does not converge well with equivalent factors due to its inferior depth resolution. We have incorporated this point into the Methods section.

Changes made:

- A complete table of parameters is provided in the supplementary materials.
- We have added the regularization factors in the Methods section.

Figure 1

Comment 3: It's not clear from the figure whether α_{eff} refers to the collection semi-angle or convergence semi angle subtended by the probe-forming aperture. Furthermore, the cones of illumination in Figure 1(a) and 1(b) are not clearly distinguished from the cone of the effective semi-angle in Figure 1(b). Furthermore, I believe it would be more helpful if the diffraction patterns were not cropped, to demonstrate the use of dark-field information for extending the depth resolution.

We appreciate the referee for these comments on Figure 1. In response to the original use of α_{eff} , it was intended as a conceptual term to represent the boundary of 3D information transfer. To clarify this distinction, we have introduced the terms β_{MEP} and β_{TCMEP} in the revised manuscript to specifically denote the semi-angles of 3D information transfer, differentiating them from the convergence semi-angle (α) and the collection semi-angle. Additionally, we have replaced the diffraction patterns in the revised Fig. 1 with uncropped versions as suggested.

Changes made:

- The term α_{eff} has been replaced by either β_{MEP} or β_{TCMEP} throughout the revised manuscript.
- The diffraction patterns in Fig. 1 are now presented in their uncropped form.

Figure 2

Comment 4: The plot of depth resolution versus dose is very helpful. Why does the depth resolution get slightly worse for larger dose for the low-tilt cases? This suggests that the reconstructions for these data points are not optimized.

We greatly appreciate the referee's suggestion. To address this, we have carefully optimized the parameters, particularly the slice thickness, for each TCMEP reconstruction. The slice thickness was chosen to be less than one third of the depth resolution to ensure consistency. Consequently, the plot of resolution versus dose, shown in Fig. R10, is now smoother and exhibits saturation at higher doses.

Fig. R10| Revised depth resolution versus total dose.

Changes made:

- Fig. R10 is used to replace the original Fig. 2f.
- The choice of slice thickness in Methods is revised accordingly.

Comment 5: Figure 2d – the y axis should emphasize that the phase is normalized. On this point, how do the phase amplitude values change with tilt before normalization?

We thank the referee for the question. We have indeed normalized the curves by their peak values, though this process is not necessary. The raw depth-sectioning curves exhibit the same trend, as shown in Fig. R11 below, which has been used for the revised Fig. 2.

Fig. R11| Depth sectioning curves of a single Sr dopant without normalization.

Changes made:

- Figure 2g is revised to display raw curves without normalization.

Comment 6: For the simulations, there is no mention of whether lattice vibrations were accounted for by using frozen phonons to model the potential. This may have a significant effect on the depth resolution and precision achievable. Furthermore, did the authors account for partial spatial and temporal coherence? This is not necessarily crucial as MEP can account for these, but I believe it is worth mentioning in a revised manuscript.

We used a frozen-phonon model with 40 configurations to simulate lattice vibrations. Spatial and temporal incoherence were not considered, as the simulations were aimed at demonstrating qualitative trends.

Changes made:

- More details are provided in the revised *4D-STEM simulation* section in Methods:

Lattice vibrations were simulated by a frozen-phonon model with 40 configurations...

Spatial and temporal incoherence was not considered since the simulations were only used to demonstrate qualitative tendencies.

Figure 3

Comment 7: The detailed analysis of experimental depth resolution is generally well-presented and impressive. However, it is not convincing that a linear fit in Figure 3k is the best method to calculate the effective semi angle. I would anticipate a quadratic relationship (which is actually visible for one of the curves) between k_r and k_z , in a similar fashion as reported in [1-3]. Can the authors comment on this? Furthermore, with this in mind, does the effective semi-angle account for the curvature of the Ewald sphere? Some important references are below:

- *Optics Express Vol. 24, Issue 25, pp. 29089-29108 (2016)*
- *Nature volume 463, pages 214–217 (2010)*
- *Journal of the Optical Society of America A, Vol. 29, Issue 8, pp. 1606-1614 (2012)*

We appreciate the referee's insightful comments. The linear fit was intended solely to highlight the slope of the k_r - k_z curve near the origin, utilizing the relatively low-frequency points. This approach is conventionally employed to determine the wedge angle in STEM optical depth sectioning methods (e.g., *Journal of Electron Microscopy* **58**, 157 (2009)). We admit that the original manuscript did not clearly convey this point, but we have never assumed a linear relationship between k_r and k_z . To clarify, we have generated k_r - k_z plots for the simulated datasets (Fig. R3a in response to Referee 1) and obtained the effective semi-angles $\beta_{\text{MEP/TCMEP}}$ through quadratic curve fitting. Fig. 3 is revised to only demonstrate the universally improved depth resolution across all lateral spatial frequencies.

In response to the questions regarding the Ewald sphere curvature, the recommended references are helpful. We would also like to highlight a reference *Phys. Rev. B* **91**, 214114 (2015), which discusses the relationship between the Ewald spheres in untilted and tilted arrangements in detail. Fig. R12 is adopted from the reference study, where Θ_{max} is the maximum scattering angle, and ω is the tilt angle relative to the y axis. The improved resolution

along the z axis (a larger $q_z' > q_z$) is expected in the single tilted geometry, whereas the resolution along the y axis degrades (a smaller $q_y' < q_y$). Fortunately, TCMEP reconstruction from multiple datasets will merge this information, and obtain a final result with expanded information up to frequencies determined by q_z' and q_y . This is essentially the same as we have illustrated in Fig. 1b.

[figure redacted]

Fig. R12 | Relationship between Ewald spheres in untilted (thin line) and tilted (bold line) arrangements. This figure is adopted from *Phys. Rev. B* 91, 214114 (2015).

Changes made:

- k_r - k_z plot for the simulated dataset is provided in Fig. 2h. Fig. 3e is revised to highlight the improved depth resolution across all lateral spatial frequencies. The main text is also revised.
- A brief discussion about the Ewald sphere curvature is provided in the *Principle and reconstruction process* section:

Such an improvement can also be interpreted through the concept of a tilted Ewald sphere as an alternative perspective²⁷.

- The suggested references were included in appropriate positions.

Comment 8: I was initially confused by the use of α_{eff} and $\alpha_{eff} + \theta$. In the current definition, $\alpha_{eff_TCMEP} = \alpha_{eff_MEP} + \theta$, but there is no use of the MEP or TCMEP subscript. The authors should add a subscript or at least refer to α_{eff} , when referring to the effective semi-angle of TCMEP reconstructions.

We appreciate the referee for these helpful suggestions. As mentioned in our response to *Comment 3*, the terms have been replaced with either β_{MEP} or β_{TCMEP} to improve clarity. Additionally, the linear relationship between β_{MEP} and β_{TCMEP} ($\beta_{TCMEP} = \beta_{MEP} + \theta$) does not accurately reflect the case in reconstructions, so we have removed this equation from the revised manuscript.

Comment 9: I believe Fourier transforms are needed to demonstrate the presence/absence of individual flakes. The current version of the figure leaves much to interpretation. For example, at a depth of 3.2 nm, I would argue that the Moiré pattern is more visible for the TCMEP reconstruction versus the MEP reconstruction. Is this because MEP has a better depth resolution at the top surface of the sample, or is it because the z-position of the reconstructions are not perfectly aligned for MEP and TCMEP? Fourier transforms for different slices in the main or supplementary figures would help clarify this point.

We appreciate the referee's valuable suggestion. We have incorporated the corresponding FFT images into Fig. 3, where the improved layer resolution is more clearly illustrated. The dashed circles in the FFT images highlight that, for both reconstructions, only the top SrTiO₃ layer is resolved in the first slice (depth = 4.8 nm). However, with MEP, the appearance and disappearance of the Moiré patterns in the depth direction occur more slowly compared to TCMEP. In the final slice, the residual Moiré pattern is still present in the MEP reconstruction, whereas TCMEP resolves only the bottom SrTiO₃ layer, demonstrating superior separation between the layers.

Fig. R13| Real-space versus FFT images for twisted bilayer SrTiO₃. The FFT images in the insets clearly demonstrate the improved layer resolution, especially with lattice peaks highlighted with the dashed circles. Scale bars are 1 nm in real space and 4 Å⁻¹ in the Fourier space.

Changes made:

- Fig. 3 is revised as described above, and the corresponding main text is also changed according to the new figure.

Figures 3a and 3b present real-space images and their corresponding Fourier transforms (FFTs), showing five slices (each 4 Å thick) from two reconstructions using MEP and TCMEP (see Supplementary Figs. 3-4 for all slices). In the first slice, only the top SrTiO₃ layer is resolved in both reconstructions, as indicated by the dashed circles in FFTs. A key distinction lies in the Moiré pattern's extent along the depth dimension. In the TCMEP results, the Moiré pattern appears and fades more rapidly along the z-axis compared to MEP, where it diminishes more slowly. In the final slice, the residual Moiré pattern is still present in the MEP reconstruction, whereas TCMEP resolves only the bottom SrTiO₃ layer, demonstrating superior separation between the layers.

Comment 10: What is the angular sampling and maximum diffraction angle in the experimental data, and how does this compare to the effective semi-angle? The angular sampling is provided for the simulated data, but there is little discussion about the maximum diffraction angle.

We appreciate the referee for raising this important question. As mentioned in response to *Comment 2*, these critical experimental parameters are provided in Supplementary Table 1. The maximum diffraction angle was 40.3 mrad (0.650 mrad/pixel) for twisted bilayer SrTiO₃, and 52.4 mrad (0.845 mrad/pixel) for (Pr,Ca)₂Co₂O₅ film. Additionally, the effective semi-angle ($\beta_{\text{MEP/TCMEP}}$) for 3D information transfer is influenced by both the maximum diffraction angle (α_{max}) and the convergence angle (α_{conv}), as discussed in a recent preprint (arXiv:2407.18063). While the ultimate limit of depth information is determined by α_{max} , the majority of 3D information transfer lies within the weak-scattering limit, which is also governed by α_{conv} . Therefore, under finite—and particularly low—dose conditions, the practical limits of information transfer are significantly affected by the probe convergence angle.

Changes made:

- The angular sampling and maximum diffraction angle have been provided in Supplementary Table 1.
- A discussion about the relationship between $\beta_{\text{MEP/TCMEP}}$, α_{conv} , and α_{max} is provided in the *Principle and reconstruction process* section:

To make a direct comparison with conventional focal-series ADF imaging, the 3D information transfer via MEP from a single dataset is qualitatively modeled as a cone with an effective semi-angle, β_{MEP} . This angle generally depends on the maximum diffraction angle, but is also constrained by the probe's convergence angle under finite—and particularly low—dose conditions^{16,19,28,29}.

Comment 11: Why is a resolution value of 1.33σ chosen instead of the FW80M ($\sim 1.28\sigma$) reported in previous works? Currently, there are inconsistencies across the field of MEP for resolution measurements, so it would be valuable to readers if the authors explained why they used this definition.

The resolution defined by 1.33σ is consistent with previous works (e.g. *Science* 372, 826 (2021)), and is based on the full-width at 80% of the maximum (FW80M) of the point spread function, approximately a Gaussian curve. We also note that a recent study (*Phys. Rev. Appl.* 22, 014016 (2024)) defines depth resolution as the difference between 10% and 90% of the maximum phase at the sample edge, leading to a value of $2 \times 1.28\sigma$ as the referee suggested, assuming the step edge is modeled using a Gaussian error function. In our simulations involving a single dopant, we used the 1.33σ definition.

Changes made:

- Definition of depth resolution is provided in detail in the *Simulation on imaging a single dopant section*:

..., determining depth resolution by the **full width at 80% of the maximum (FW80M, $d=1.33\sigma$)**, following the method in prior work².

Comment 12: How exactly are the error functions determined from the curve fitting? I think you need to specify 'the residuals of the curve fitting' here (if this is what was actually done).

We appreciate the referee for this valuable comment. As the referee assumed, the error bars are based on the residuals from curve fitting. We have revised the figure captions to clarify this point.

Changes made:

- We specify the determination of error bars in the captions of figures:

Error bars are derived from residuals of curve fitting.

Figures 4 and 5:

Comment 13: How can you decouple lattice distortions from scan position errors? I think the authors solve this problem via registration of the scan positions for each tilted data set, but this should be clarified to aid the reader

We greatly appreciate the referee's question regarding probe-position correction. As the referee correctly assumed, we register the scan positions for each tilted dataset and apply position correction individually to each one.

Changes made:

- An additional supplementary figure (Supplementary Fig. 13) is provided to illustrate the detailed TCMEP reconstruction results including probes, positions and tilt angles.

Comment 14: Lines 127-129: “On the low dose side with comparable tilt angles, the improvement of TCMEP is more significant, reaching a depth resolution of around 0.55 nm, which is more than 4 times better than that achieved with MEP”. This is an impressive result. It seems related to overlapping bright-field disk data between adjacent tilts, as low-dose single tilt data would have little dark field signal above the noise level. Can you comment on the dose efficiency of multiple bright-field tilts versus fewer dark field tilts? No further data/processing is necessary, but a commentary would be appreciated by readers.

We sincerely appreciate the referee’s insightful question. Upon reviewing the low-dose simulated datasets, we confirmed that only the bright-field portions of the diffraction patterns contain significant counts above the Poisson noise level. We agree with the referee that the improvement in depth resolution under low-dose conditions is primarily due to the bright-field information coupled with sample tilts, rather than contributions from dark-field regions.

Changes made:

- We have added a discussion on the origin of dose efficiency in TCMEP:

This improvement primarily results from the bright-field disk where electrons contribute significantly, while dark-field regions remain below the Poisson noise level at low doses. TCMEP incorporates higher-angle information into the bright-field disk through specimen tilts, thereby significantly improving dose efficiency in depth sectioning.

Other

Comment 15: The dashed rectangles in Figure 5(c) are difficult to see and should have thicker lines in a revised manuscript

We appreciate the referee’s valuable suggestion. In response to referee #1’s comments, we have completely revised Figure 5. We have carefully selected appropriate line widths and colors to improve the clarity of all the new figures.

Comment 16: Line 285: Extended Fig 7A – this should be Extended Fig 6A

We thank the referee for pointing out the typo. It has been corrected in the revised manuscript.

Comment 17: Ref. 18 – A quick Google search found a published result for this work: Phys. Rev. Applied 22, 014016 (2024)

We appreciate the referee’s suggestion. The reference has been updated to the published version as recommended.

Comment 18: Line 76-77: At a mild illumination intensity

The sentence has been revised according to the suggestion.

Comment 19: Line 223-224: “Despite a slight sacrifice in depth resolution, the results demonstrate much better overall quality and precision.” – the authors should cite the paper which demonstrates poorer precision than the results shown in this manuscript. Furthermore, “better overall quality” is quite vague. Please replace with a quantitative comparison and a relevant citation to the literature.

We appreciate the referee’s suggestion. The sentence has been revised as recommended, and a reference has been attached as follows:

Despite a slight sacrifice in depth resolution (compared to ref.⁴⁸), our results demonstrate **sub-angstrom lateral resolution and around one-picometer precision, along with a marked improvement in sensitivity to light atoms.**

Comment 20: A spelling and grammar check should be performed before submitting a revised manuscript version.

We appreciate the referee’s suggestion. We have thoroughly reviewed the manuscript and corrected all identified grammatical and spelling errors. We believe these revisions have enhanced the manuscript’s readability.

Response to Referee: 4

As a atom sized electron probe transits a specimen it picks up a rich variety about the atomic structure of the target. It has long been recognized within the electron microscopy community that this information is encoded within the electron probe, whilst detailed, is difficult to reconstruct due to the strong multiple scattering of the illumination. There has been recent progress in answering this research question due to the advent of 4D-STEM, where modern cameras record the full 2D diffraction pattern from a focused electron at a speed commensurate with the 1-100 microsecond timescales of the 2D probe scan, and newer reconstruction algorithms. Multislice electron ptychography has thus far proven the most successful algorithm for turning 4D-STEM data into reconstructions of the sample, with demonstrable improvements in resolution once multiple scattering artifacts are successfully removed from data. While resolution improvements are an impressive demonstration of the power of these new algorithms the ability to reconstruct more 3D information (which is usually lost in a typical transmission electron microscope experiment) about the sample is a far more illustrious goal and one that seems to have been achieved with this publication. Whilst 3D information is often reconstructed in electron microscopy using a tomography tilt series these are challenging for slab-like crystalline specimens since electron multiple scattering introduces a strong, non-linear effect on image contrast as a function of tilt. In this work a small tilt (2-4 degrees) is shown to be sufficient to vastly improve the robustness of multislice ptychography such that single atom dopant detection and the resolution of separate layers in a heterostructure are now much better detected. Overall the paper is well written and structured, the demonstrations of the technique on simulated and experimental datasets are well presented and the results are impressive and have the potential to have a large impact on the TEM field going forward. I recommend publication subject to a few minor changes.

We greatly appreciate the referee's insightful summary of the evolution of 4D-STEM techniques and electron ptychography algorithms, which highlights the novelty of our manuscript. We agree with the referee and also notice the recent huge impact of multislice electron ptychography in electron microscopy community. Below, we address each of the referee's comments point-by-point.

Introduction

"Real-space imaging of three-dimensional atomic structures is a critical yet challenging task in materials science" would be more logically worded as "Real-space, three-dimensional imaging of atomic structures in materials science is a critical yet challenging task"

We appreciate the referee's suggestion. The sentence has been revised as recommended to improve its readability.

"This technique requires only a moderate level of data acquisition or processing, and can be seamlessly integrated into electron microscopes equipped with conventional components." -> This statement does not strike me as factual, 30 hours of processing on an A100 GPU is hardly moderate and the technique requires at least a high end microscope equipped with a FEG, probe aberration corrector and hybrid pixel detector some of which are not considered

"conventional components" yet. Not state of the art specialist equipment but not what will be found at many TEM centers. The statement: "The technique can be applied on a high-end, commercially available TEM with a probe aberration corrector and hybrid pixel detector and data processing is accessible to modern high-performance computing systems." is more accurate.

We agree with the referee that TCMEP reconstruction is indeed a complex technique requiring advanced hardware. To reflect this, we have revised the abstract sentence as follows:

Our approach can be implemented on widely available transmission electron microscopes equipped with hybrid pixel detectors, with data processing achievable using high-performance computing systems.

Principal and reconstruction process

*The theoretical explanation of using 3D linear contrast transfer functions to my mind lacks insight. The improvement in depth resolution cannot be explained with linear imaging theory and this is readily seen by a back of the envelope calculation. To a first approximation sufficient for discussion here, a single image in STEM gives information in Fourier space in wedge equal to the probe forming semi-angle. For the 25 mrad semi-angle used in this work this is corresponds to 1.4 degrees, so the missing wedge would be $180 - 2 \times 1.4 = 177.2$ degrees! If the sample is tilted by 4 degrees this would missing wedge would reduce to a still pitiful 169.6 degrees which would still be highly unsatisfactory. The actual results in this paper suggest a far more impressive improvement with small tilts than the theoretical analysis (which should give an upper bound in improvements) would suggest and to my mind this is better explained through better diversity of inputs into the phase retrieval algorithm. A hallmark of strong multiple electron scattering by crystalline samples is the strong dependence of TEM results on even small (few mrad) tilts of the specimen so different tilts will give a much more varied input to the (famously unstable without strong regularization see Schloz, Marcel, et al. "Overcoming information reduced data and experimentally uncertain parameters in ptychography with regularized optimization." *Optics Express* 28.19 (2020): 28306-28323.) multislice ptychography algorithm. This improved and more diverse input should result in a much more reliable and more accurate reconstruction of the object function which is what the authors observe. This hypothesis might be tested by comparing ptychography results with and without the small tilt for an object that doesn't give strong multiple scattering, eg. a very small (few nm) crystal nanoparticle, to see if a similarly impressive improvements in depth resolution are realized.*

We sincerely appreciate the referee's insightful comments. We recognize that the initial linear relationship proposed between wedge angle and tilt angle, $\beta_{TCMEP} = \beta_{MEP} + \theta$, was indeed an oversimplification, and we have removed this linear assumption in the revised manuscript.

We also acknowledge the referee's valuable suggestion to perform TCMEP reconstructions on simulated datasets for a weakly-scattering SrTiO₃ nanoparticle with a 4×4×4 nm³ volume (Fig. R14a), while maintaining a constant illumination dose of 2.5×10⁶ e/Å² (same as Fig. 2a-h). The results confirm the referee's assumption that depth resolution is indeed improved compared to the results in Fig. 2. We observe depth resolutions of 0.73, 0.59, and

0.41 nm (Fig. R14e), which is undoubtedly better than that of 1.20, 0.73, and 0.50 nm in the 10-nm thick crystal under the same dose conditions (Fig. 2j). Therefore, we agree with the referee that in the weak-scattering regime, the CBED patterns are encoded with phase information that is much easier to retrieve (Fig. R14b-c), which can lead to better depth resolution. However, incorporating tilt series can still improve depth resolution even within this weak-scattering context, although the improvement is not as significant as that in the strong-scattering regime (Fig. R14d-e). In summary, TCMEP consistently improves depth resolution across different scattering regimes, and the improvement is more significant in the strong-scattering regime.

Fig. R14| TCMEP results on a simulated weak-scattering SrTiO₃ nanoparticle. **a**, Projected phase image of the SrTiO₃ nanoparticle (4×4×4 nm³), reconstructed using MEP. **b-c**, Convergent beam electron diffraction (CBED) patterns for SrTiO₃ nanoparticle (**b**) and SrTiO₃ crystal in the main text (**c**), corresponding to the weak-scattering regime (**b**) and the strong-scattering regime (**c**) respectively. **d**, Upper panels: reconstructed phase images cropped into the yellow dashed regions in panel **a**, using maximum tilt angles of 0° (left), 2° (middle) and 4° (right). Lower panels: depth profiles along the arrows in the upper panels. The total illumination dose is 2.5×10⁶ e/Å². **e**, Phase-depth curves for the Sr dopant. The depth resolutions are 0.73 nm, 0.59 nm, and 0.41 nm, respectively.

Changes made:

- We avoid the usage of the linear assumption of $\beta_{\text{TCMEP}} = \beta_{\text{MEP}} + \theta$ throughout the manuscript.
- We added Fig. R14 as Supplementary Fig. 9 and provided a corresponding discussion about scattering strength in the *Discussion* section.

Depth resolution in MEP-based techniques is also influenced by the efficiency in retrieving phase information from convergent-beam diffraction patterns. For example, MEP reconstruction on simulated datasets for a weakly scattering SrTiO₃ nanoparticle (Supplementary Fig. 9) demonstrates a superior depth resolution (0.73 nm) compared to that for a strongly scattering SrTiO₃ crystal (Fig. 2) under identical parameters (1.20 nm). This improvement is primarily due to the more interpretable diffraction patterns produced by nanoparticles, where multiple scattering effects are significantly reduced (Supplementary Fig. 9b-c), facilitating three-dimensional phase retrieval. Notably, even in this weak-scattering context, TCMEP enhances depth resolution from 0.73 nm (maximum tilt 0°) to 0.41 nm (maximum tilt 4°), indicating TCMEP's broad applicability across different scattering regimes.

In the TCMEP reconstruction algorithm are the different planes of the object truly shifted in real-space or does the algorithm incorporate a tilted specimen by a tilted Fresnel propagation function (ie. shifted in reciprocal space) as a standard multislice calculation would account for tilts? The latter strikes me as more accurate though the former might be serviceable.

For our work, we employed a shifted object function in TCMEP reconstructions, which met our needs. However, we acknowledge that using a tilted Fresnel propagator is an alternative way to simulating small sample tilts. Furthermore, incorporating tilt-angle corrections would be more straightforward with the tilted Fresnel propagator, as suggested by Sha et al. (Sci. Adv. 8, eabn2275 (2022)).

Changes made:

- We have added a sentence in the *Methods-Alignment* section, which describes the object function shifting in our work and discusses the tilted Fresnel propagator as an alternative approach.

In the TCMEP reconstruction process, sample tilts were modeled by shifting the object functions, although a tilted Fresnel propagation function could serve as an alternative approach⁵⁰.

Perhaps state clearly that the tilts used a specimen (ie. stage) tilts, beam tilts are an alternative though this will likely induce aberrations which must then be solved jointly in the reconstruction.

We greatly appreciate the referee's suggestions. In the revised manuscript, we have emphasized the role of specimen tilt and included a brief discussion on beam tilt, outlining its associated challenges in TCMEP experiments and reconstructions.

Changes made:

- Beam tilt is discussed as an alternative in the *principle and reconstruction process* section: These datasets are acquired from the same region of the sample, with the specimen intentionally tilted by small angles—significantly less than 1 radian—away from the zone axis (Fig. 1a). ...

While beam tilt can serve as an alternative to specimen tilt, it introduces additional aberrations in the electron beam, complicating TCMEP reconstructions.

Simulation on imaging with a single dopant

Could the authors please include an image of the depth section of the dopant atom for the different maximum tilts in Fig. 2? Even for high quality electron tomography work these give an unflattering image of the quality of reconstruction but can reveal the extend of the missing wedge.

We sincerely appreciate the referee's excellent suggestion to include depth-sectioning images. We have now added Fig. R15, corresponding to the simulation results presented in Fig. 2. This addition significantly enhances the clarity of demonstrating the improved depth resolution achieved through TCMEP reconstructions.

Fig. R15| Depth sectioning images of TCMEP simulation results. a-c, Slice images of a single Sr dopant for reconstructions with maximum tilt angles of 0° (a), 2° (b) and 4° (c). **d-f,** Depth sectional images along the broken arrows in panels a-c, respectively.

Changes made:

- Fig. R15 is included in the revised Fig. 2.

Depth resolution to sub-nanometers

Could the authors include inset images of the Fourier transform of each reconstructed slice in Fig 3? This will give a better visual indication of layer resolution. The atomic model from vesta in 3 (i) could also be tweaked by deleting the atoms from each layer of the moire stack for some fraction of the supercell so that the individual SrTiO₃ lattices can be seen and how they form a Moire in projection.

We appreciate the referee's valuable suggestions. We have incorporated the corresponding FFT images into Fig. 3, which now clearly illustrate the improved layer resolution. The dashed circles in the FFT images highlight that, for both reconstructions, only the top SrTiO₃ layer is resolved in the first slice (depth = 4.8 nm). However, with MEP, the appearance and disappearance of the Moiré patterns in the depth direction occur more slowly compared to TCMEP. In the final slice, the residual Moiré pattern remains in the MEP reconstruction, whereas TCMEP resolves only the bottom SrTiO₃ layer, demonstrating superior separation between the layers.

Additionally, we have revised the atomic model for twisted bilayer SrTiO₃, following the referee's recommendations, to better display the individual layers along with the Moiré pattern formed at the interface.

Fig. R16| Atomic model and FFT images for twisted bilayer SrTiO₃. **a-b**, FFT images in the insets clearly demonstrate the improved layer resolution, especially with lattice peaks highlighted with the dashed circles. Scale bars are 1 nm in real space and 1 Å⁻¹ in the Fourier space. **c**, Atomic model showing individual layers and the Moiré pattern formed at the interface.

Changes made:

- Fig. 3 is revised as described above, and the corresponding main text is also changed according to the new figure.

Figures 3a and 3b present real-space images and their corresponding Fourier transforms (FFTs), showing five slices (each 4 Å thick) from two reconstructions using MEP and TCMEP (see Supplementary Figs. 3-4 for all slices). In the first slice, only the top SrTiO₃ layer is resolved in both reconstructions, as indicated by the dashed circles in FFTs. A key distinction lies in the Moiré pattern's extent along the depth dimension. In the TCMEP results, the Moiré pattern

appears and fades more rapidly along the z -axis compared to MEP, where it diminishes more slowly. In the final slice, the residual Moiré pattern is still present in the MEP reconstruction, whereas TCMEP resolves only the bottom SrTiO₃ layer, demonstrating superior separation between the layers.

I disagree with the analysis presented in Fig. 3 (k), the fitted straight lines (dashed lines) do not match the experimental measured profiles of k_r vs k_z well at all since the straight line fit is constrained to start at the origin so I don't believe the estimates of missing wedge that flow from this analysis. I think this part should be removed from the paper. The same applies to Fig. 5c of the supplementary.

We appreciate the referee's insightful comment. The linear fit was intended solely to highlight the slope of the k_r - k_z curve *near the origin*, utilizing the relatively low-frequency points. This approach is conventionally employed to determine the wedge angle in STEM optical depth sectioning methods (e.g., *Journal of Electron Microscopy* **58**, 157 (2009)). We admit that the original manuscript did not clearly convey this point, but we have never assumed a linear relationship between k_r and k_z .

To clarify, we have generated k_r - k_z plots for the simulated datasets (Fig. R3a in response to Referee 1) and obtained the effective semi-angles $\beta_{\text{MEP/TCMEP}}$ through quadratic curve fitting. We have also revised the experimental Fig. 3 to only demonstrate the universally improved depth resolution across all lateral spatial frequencies.

Changes made:

- Fig. 3e is revised, while a new Fig. 2h is provided.
- We have provided a corresponding discussion paragraph on Fig. 2h for simulated datasets as follows:

The Fourier transform of the reconstructed phase image reveals the boundary of information transfer¹⁹ (Fig. 2h, details in Supplementary Fig. 2), which qualitatively agrees with the schematic illustration in Fig. 1b, showing that the boundary expands along the depth dimension across all lateral spatial frequencies. The effective semi-angles $\beta_{\text{MEP/TCMEP}}$, defined by the slopes of fitted quadratic curves near the origin, are measured at 207 mrad (11.8°), 352 mrad (20.1°), and 553 mrad (31.6°), respectively. Notably, β_{MEP} in our simulation is around three times larger than the corresponding value from experimental results reported in ref.¹⁹. This discrepancy primarily arises from the idealized simulation conditions, which do not account for experimental uncertainties such as sample drift or partial spatial-temporal coherence. Moreover, the unavoidable roughness of sample surface broadens the depth distribution in the Fourier spectra, leading to an overestimation of depth resolution in the experimental results. Nevertheless, it has been confirmed that the improvement in depth resolution is primarily driven by information gathered at higher angles.

- We have provided a discussion paragraph on Fig. 3e for experimental datasets as follows:

Moreover, the corresponding Fourier analysis (Fig. 3e, details in Supplementary Fig. 6) reveals a universally improved depth resolution across all lateral spatial frequencies as the maximum tilt angle increases, consistent with the findings in Fig. 2h for the simulated datasets.

Alignment of different tilts and the reconstruction process

This step seems critical to the whole algorithm working well and I'm impressed that the approach outlined in this section has worked as well as it has.

For one of the experiments, eg. Fig 3, could the authors please present as a supplementary figure showing the alignment of the separate ptychographic reconstructions for each acquisition in the tilt series and a plot showing the change in refined probe positions from the drift-correction algorithm built into the ptychographic reconstruction?

We appreciate the referee's request for more details on data alignment and position correction. Indeed, proper alignment is crucial for successfully reconstructing multiple 4D-STEM datasets using the TCMEP approach. Both experiments presented in the main text follow similar experimental procedures and numerical refinements. For clarity, we have now included alignment details for $(\text{Pr,Ca})_2\text{Co}_2\text{O}_5$ in Fig. R17 and position refinements for twisted SrTiO_3 in Fig. R18. The processes for both datasets are generally the same.

Fig. R17| 4D-STEM data alignment prior to TCMEP reconstruction. **a-c**, MEP reconstruction results for the full datasets acquired at tilt angles of 0° (**a**), -1° (**b**) and $+1^\circ$ (**c**). The intentional electron-beam-induced defect area is used as a reference for alignment, with the region of interest (ROI) highlighted by a blue square. **d-f**, MEP reconstruction results for the cropped ROI within the datasets acquired at tilt angles of 0° (**d**), -1° (**e**) and $+1^\circ$ (**f**). The results demonstrate good alignment and are used to initialize the TCMEP reconstruction, while the slight residual misalignments will be further refined during the iterations.

Fig. R18| Position-correction results for TCMEP. Refined scan positions for each dataset, while position corrections are performed individually to each of them.

Changes made:

- Fig. R17 (alignment of datasets) is provided as Supplementary Fig. 12.
- An additional supplementary figure (Supplementary Fig. 13) is provided to illustrate the detailed TCMEP reconstruction results including probes, positions and tilt angles.

These citations should be added to the manuscript due to the strong overlap of the work.

Schloz, Marcel, et al. "Improved Three-Dimensional Reconstructions in Electron Ptychography through Defocus Series Measurements." *arXiv preprint arXiv:2406.01141* (2024).

Ren, David, et al. "A multiple scattering algorithm for three dimensional phase contrast atomic electron tomography." *Ultramicroscopy* 208 (2020): 112860.

We are grateful to the referee for suggesting relevant citations that were missed in the original manuscript. These have now been incorporated into the revised version.

Changes made:

- The first reference appeared around the same time as our submission, so we included a citation in a *Note added in proof* section after the main text.

Note added in proof: During the review process of this manuscript, we noticed another work by Schloz et al.⁴⁹, which proposed a defocus-series strategy to improve the three-dimensional reconstruction of multislice electron ptychography.

- The second reference is added to the following sentence in the 2nd paragraph in the *Discussion* section:

Notably, TCMEP can be extended to larger tilt angles with the implementation of projection algorithms used in tomography, as demonstrated by previous simulation studies⁴³⁻⁴⁶.

Response to Referee: 1

We sincerely thank the reviewer again for their detailed feedback on our manuscript and for pointing out areas requiring further clarification, although some of the comments were quite confusing. We address the comments below, aiming to ensure that the manuscript provides an accurate and clear presentation of our work.

The authors have responded satisfactorily to many of the detailed remarks in the previous comment. However, the main criticism was that they have claimed to having developed a novel reconstruction principle (TCMEP) and in their rebuttal they write: “we would like to highlight that the artificial neural network (ANN) framework used in Refs. [1,2] differs from the tilt-coupled multislice electron ptychography (TCMEP) utilized in our study.”

I ask the authors to explicitly state the difference. While the new references [25,26] in the manuscript mention the analogy to ANN in their manuscript before describing explicitly the analytical expressions for the gradients they use in their CG optimization, reference [54] cited by the authors for the LSQML principle applied by them provides expressions for the gradients without referring to ANNs and also apply CG optimization. While ref. [25] uses a loss-function in form of the sum of squared differences, ref. [57], which the corresponding author is co-author on, compares different loss functions, including the log-likelihood that is also the basis for the LSQML optimization used in this manuscript. So, I would say that in terms of the reconstruction principle, the authors have reimplemented and applied already published algorithms. This should be made very clear in the manuscript. In their revised version, the authors write already in the abstract “Here, we introduce a new algorithm based on multislice electron ptychography, ...” – In light of the above argument, this is very deceiving. Also other places, they write “Our approach...” – Apart from the alignment of probe positions across data sets of different tilt (see below), others have developed this “approach” first.

We appreciate the reviewer’s request for explicit clarification regarding the differences between TCMEP and previous ANN-based works. However, we are unable to follow the logic behind the reviewer’s comment questioning the novelty of our work. As clearly stated in our manuscript, TCMEP represents a further extension of multislice electron ptychography (MEP): “Here, we introduce a new algorithm **based on** multislice electron ptychography”. At no point have we claimed that all the algorithms used in this manuscript for MEP or TCMEP were solely developed by us. As the reviewer also acknowledged, even in earlier works coauthored by the corresponding author (e.g., Ref. [16]), we have never asserted the invention of the fundamental principles of MEP; instead, we have included the necessary references. In fact, solving such problems fundamentally requires employing gradient descent algorithms to optimize results based on a defined loss function. Therefore, the similarity between ANN architectures and LSQML architectures at a macroscopic level is inevitable. We also note that the suggested references have included similar statements: ‘This is not uncommon, in related fields—like electron tomography, coherent diffraction imaging, and ptychography—solutions often are arrived at iteratively.’ (Phys. Rev. Lett. 109, 245502 (2012)).

According to the reviewer’s logic, an approach can only be considered novel if its basic principles, such as conjugate gradient (CG) optimization or loss functions, are entirely different.

This is definitely not common sense. For example, in Ref. [54], the authors indeed used similar or identical expressions for gradients and CG optimization as those used in ANNs described in Refs. [25-26], but it would be unreasonable to suggest that Ref. [54] should cite ANNs, given that it was published even a few months prior to the paper that first introduced ANNs (Phys. Rev. Lett. 109, 245502 (2012)) and the subsequent works in Refs. [25-26]. If the reviewer intended to argue for the priority of ANNs over LSQML-MEP for the authors in Refs. [25, 26], such a discussion would be more appropriate in a formal publication rather than during the review process of this manuscript, as we have never engaged in such an argument.

We outline the key differences between the ANN-based work in Refs [25, 26] and the present LSQML-based methods below:

- **Position correction:** the ANN-based work did not incorporate position correction, which is a crucial step in reconstructing experimental datasets.
- **Probe reconstruction:** the ANN-based work did not include a complete probe reconstruction except from a defocus optimization, as it was applied only to simulated datasets where the electron probes were ideal and already known.
- **Treatment of partial spatial coherence:** the ANN-based work applied an extra convolution after taking the intensity, whereas the LSQML-based methods utilized mixed-state probes (Ref. [53]).
- **Regularization:** the ANN-based work included l_1 -regularization (for sparse objects), while the LSQML method employed layer regularization (critical for reconstructing experimental datasets).
- **Alignment of datasets:** the ANN-based work used simulated datasets from different tilt angles, which did not require further alignments. Accurate alignment of positions from different tilts is a prerequisite for high quality experimental realization.

We would like to emphasize that by ‘our approach’, we refer specifically to the implementation described in our manuscript. While our work builds upon previously established LSQML-MEP methods, its primary contribution lies in the development of joint tilt-series reconstructions and their successful experimental realization.

Despite the above rebuttals, we acknowledge the reviewer’s concerns and have revised certain sentences in the manuscript to improve clarity.

- The abstract sentence is revised to ‘Here, we introduce *an extension of multislice electron ptychography, ...*’.
- Also, in abstract ‘Our approach’ is revised to ‘**This approach**’.

The authors continue to write in their rebuttal: “Several critical experimental factors—such as the retrieval of the incident probe function, correction of probe position errors, and alignment across different datasets—were not addressed in Refs. [1,2], yet these are essential prerequisites for successful experimental reconstructions.” I again have to object. The update of probe position and probe wave function is standard in ptychography. Ref. [1] has a separate section in the appendix with the explicit expression for the gradient with respect to the defocus

of the probe. Ref. [57] (same authors as [1,2]) devotes also a detailed analysis to the refinement of the probe positions and full probe wave function(see e.g. Figs. 3 & 4 in ref. [57] and all of section 3). Also the alignment across different data sets of different defocus has been done by the same authors of in ref. [49].

We are somewhat perplexed by the referee's comments. The referee suggests that position corrections, probe reconstructions, and data alignments are standard practices in ptychography, yet the suggested ANN-based references do not appear to implement these crucial elements. It is widely recognized that practical reconstructions of experimental data can be significantly affected by the implementation of these corrections, even though the basic principles are common.

Regarding probe reconstruction, optimizing the defocus alone does not contribute to accurate reconstruction of the actual probe function. With respect to position refinements, Ref. [57] did not incorporate such refinements for tilt-series reconstructions, nor did it address the challenges discussed in the ANN-based references. Concerning data alignment, Ref. [49] was only recently available on arXiv, after the initial submission of our manuscript. Furthermore, while they used simultaneously acquired annular dark-field images to align datasets with varying defocus values, they did not integrate this alignment into the reconstruction algorithm, which is a critical aspect of our work as described in this manuscript.

Here is one more example (exemplary for other places in ms) of where I disagree with the authors interpretation of ref. [25]. They write: "Simulations show that the most significant improvement occur at small tilt angles, with atomic-scale depth resolution achievable at tilt angles of approximately 4°, corroborating previous proof-of-principle demonstrations using similar reconstruction schemes [25,26]."

We are unsure why the reviewer disagreed with our interpretation of Ref. [25]. Unlike Refs. [25,26], which did not conduct comprehensive simulations addressing factors such as dose and depth resolution, we have performed systematic simulations under practical experimental conditions. Our conclusions, supported by quantitative analyses, are summarized in Fig. 2, providing a more thorough perspective compared to the proof-of-principle demonstrations in Refs [25,26].

I agree with the authors (and have stated this in my report on the first version) that this is the first experimental implementation of including multiple tilts in the reconstruction. However, apart from the refinement of tilt angles for each dataset and the alignment of probe positions between data sets of different tilt, I do not at all agree with the argument of novelty by the authors.

We have addressed the referee's comments. In summary, the suggested early ANN-based approaches, in their published forms, were not directly applicable to experimental datasets. These methods lacked essential components such as probe reconstruction, position refinement, and dataset alignment, which are critical even for basic MEP reconstructions of experimental data without tilting. We hope these revisions adequately address the reviewer's concerns and

provide greater clarity on the contributions and context of our work within the broader field of ptychographic reconstruction.

In addition, we have made our code publicly available on Zenodo. We believe that TCMEP has the potential to inspire numerous new applications and address key challenges in materials science.

Response to Referee: 2

We greatly appreciate the collaborative effort of the referee in reviewing the manuscript and will take into account all the feedback provided.

Response to Referee: 3

Remarks to the Author:

I want to thank Dong et al. for comprehensively addressing all of my comments from the previous review. The paper reads well and explains the research such that it can be reproduced by others. I recommend this article for publication. I have a few minor comments that could be addressed to improve the article quality:

We sincerely thank the referee for their positive assessment and recommendation for publication. We also appreciate the comments provided and have carefully addressed them point-by-point to further enhance the quality of our article.

- The use of the words 'remarkable' and 'unprecedented' are not valid here, as I believe existing theory would suggest that this experiment could be achieved. I think the authors should focus on quantitative values to demonstrate the significance of their result. Similarly, the use of 'only moderate resolutions' for other people's work is unfair and subjective. This should be replaced with something more objective, i.e. the ratio of depth resolution vs lateral resolution.

We have followed the referee's suggestions:

- The word 'remarkable' at line 129 is revised to 'substantial': '..., both showing **substantial** improvements in depth resolution'.
- The word 'unprecedented' at line 162 is deleted: 'we achieve **a depth resolution** of approximately 0.9 nm'
- 'Moderate resolutions' at line 247 is revised as: 'reconstructions combining MEP and tomography have so far achieved **3D resolutions** around 1.75 Å and precision of 17 pm **using 36 projections with a maximum tilt angle of 63°**^{47,48}.

At line 251: As shown above, we used only 4 projections with a maximum tilt angle of 2° to achieve a depth resolution of about 9 Å and a lateral resolution of better than 0.4 Å.

- line 58: proof-of-principle is spelled incorrectly

We have corrected this typo.

- line 185: values of lateral resolution would be welcome here

We have provided corresponding estimation of lateral resolutions: 'both achieving similar lateral resolution **at around 0.4 Å**'

Remarks on code availability:

There was no code provided.

We have made the repository of our codes and datasets available on Zenodo (Ref. [59]), accompanied by a README file with instructions for code execution.

Response to Referee: 4

Remarks to the Author:

To my reading this is a sound piece of research and the authors have comprehensively addressed some of the suggestions I have made in my previous review. I recommend publication of the work.

We sincerely thank the referee for their positive assessment and recommendation for publication.

Remarks on code availability:

The code is listed as "available upon reasonable request", best practice is to upload the code to Github and to create a release on Zenodo

We have made the repository of our codes and datasets available on Zenodo (Ref. [59]), accompanied by a README file with instructions for code execution.